# Evanescent scattering imaging of single protein binding kinetics and DNA conformation changes

Pengfei Zhang [1,5], Lei Zhou [2,5], Rui Wang [1], Xinyu Zhou [1,3], Jiapei Jiang [1,3], Zijian Wan [1,4] & Shaopeng Wang [1,3✉]

Evanescent illumination has been widely used to detect single biological macromolecules because it can notably enhance light-analyte interaction. However, the current evanescent single-molecule detection system usually requires specially designed microspheres or nanomaterials. Here we show that single protein detection and imaging can be realized on a plain glass surface by imaging the interference between the evanescent lights scattered by the single proteins and by the natural roughness of the cover glass. This allows us to quantify the sizes of single proteins, characterize the protein–antibody interactions at the single-molecule level, and analyze the heterogeneity of single protein binding behaviors. In addition, owing to the exponential distribution of evanescent field intensity, the evanescent imaging system can track the analyte axial movement with high resolution, which can be used to analyze the DNA conformation changes, providing one solution for detecting small molecules, such as microRNA. This work demonstrates a label-free single protein imaging method with ordinary consumables and may pave a road for detecting small biological molecules.

[1] Biodesign Center for Bioelectronics and Biosensors, Arizona State University, Tempe, AZ, USA. [2] Center for Biological Physics, School of Molecular Sciences, Department of Physics, Arizona State University, Tempe, AZ, USA. [3] School of Biological and Health Systems Engineering, Arizona State University, Tempe, AZ, USA. [4] School of Electrical, Energy and Computer Engineering, Arizona State University, Tempe, AZ, USA. [5] These authors contributed equally: Pengfei Zhang, Lei Zhou. ✉email: Shaopeng.Wang@asu.edu

Single-molecule detections push beyond ensemble averages and reveal the statistical distributions of molecular sizes and binding processes. Fluorescence microscopy is firstly developed and has been widely used for this purpose by shifting the detection wavelength from excitation wavelength to dramatically reduce the background for improving the signal-to-noise ratio, thus allowing single-molecule imaging[1–3]. Besides, the gold nanoparticles and chain polymers can also be used to increase the light-analyte interaction cross-section for single-molecule detection[4–7]. In contrast with above-mentioned label involved techniques, label-free single-molecule detection has been developed in the past two decades to analyze the intrinsic molecular properties, such as size and mass, along with monitoring the molecular interaction process without labels[8–11]. Evanescent illumination is usually employed for label-free single-molecule detection, because it can enhance light-analyte interaction and reduce background by notably reducing the illumination volume[10,12–14]. However, specific optical structures, such as microspheres and nanomaterials, are required to efficiently couple the incident light into the evanescent field[15]. Until now, it is still challenging to employ these exquisite microspheres and nanomaterials for wide-field single-molecule imaging applications, such as parallelly monitoring the dynamic molecular binding process in different locations[16]. Recently developed plasmonic scattering microscopy (PSM) utilizes the surface plasmonic wave propagating along the surface of the gold-coated glass slide as evanescent illumination, which notably simplified the system structure[17]. However, the plasmonic field generates much heat at high incident light power[18], limiting its applications for detecting temperature sensitive biological molecules as well as long-term monitoring of molecular interaction processes[16].

Here we show that evanescent single-molecule imaging can be successfully achieved on a plain glass surface with total internal reflection (TIR) configuration, via imaging the interference between the evanescent lights scattered by the single-molecules and by the natural roughness of the cover glass. To differentiate this imaging method from various kinds of TIR applications, we refer it to as evanescent scattering microscopy (ESM). We first describe the ESM setup and principles, and then calibrate the system with proteins of different sizes. Next, we demonstrate that ESM can analyze molecular binding kinetics and explore the heterogeneity of single protein binding properties. Finally, owing to the exponential distribution of evanescent field intensity, the ESM can track the axial movement with high resolution as surface plasmon resonance (SPR)[7]. We demonstrate that this feature can be used to analyze the DNA conformation changes, providing one solution for detecting small biological molecules, such as microRNA (miRNA). Small molecule detection is challenging for label-free optical measurement systems, which are usually considered only suitable for detecting biological macromolecules[10].

## Results

**ESM setup and imaging principles.** We excite evanescent waves by directing collimated light at an incident angle of ~65°, which is slightly larger than the critical angle of ~61.8° at BK7 optical glass-water interface, via an oil-immersion objective onto a commercial cover glass placed on the objective (Fig. 1a and Supplementary Fig. 1 for details). The incident angle was fixed in all experiments to ensure stable penetration depth of evanescent field (Supplementary Note 1). Traditional microscopic TIR imaging records the reflection beam for extinction spectral analysis or local refractive index sensing[19,20], but the strong total reflection light limits the maximum incident light intensity, making it challenging to achieve sufficient signal-to-noise ratio for single-molecule detection. In contrast, the ESM detects evanescent waves scattered by the analyte ($E_s$) using a second objective placed

on top of the sample to overcome this difficulty. Unsurprisingly, we also observe the background ($E_b$) from the evanescent waves scattered by the natural surface roughness of cover glass (Fig. 1b), similar to the phenomena observed on PSM using the gold-coated glass slides[17]. The nanometer-scale surface roughness can also be revealed by atomic force microscopy (Supplementary Fig. 2). We select the cover glasses with suitable roughness profiles from the commercial batches to ensure measurement reproducibility (Supplementary Fig. 3). The ESM image intensity of an analyte can be given by

$$I = |E_b + E_s|^2 = |E_b|^2 + |E_s|^2 + 2|E_b||E_s|cos(\theta) \qquad (1)$$

where $\theta$ is the phase difference between light scattered by analyte and surface roughness. The phase difference determines whether the interferometric contrast, namely the $2|E||E|\cos(\theta)$, is negative or positive. Unlike the interferometric scattering, where the Gouy phase shift dominates and leads to a negative interferometric contrast[11,21], the phase difference is close to zero in ESM because of the short distance between scattering sites of surface roughness and analyte binding positions, resulting in a positive interferometric contrast as the PSM[16,17,22,23]. Besides, the phase statistics are identical within the field of view, which is determined by illumination area (Fig. 1b), because the phase difference is random in the multiple scattering regime[24]. For analytes with large sizes, such as polystyrene nanoparticles with a diameter of 143.6 nm, the $|E|^2$ is much larger than $|E_b|^2$ (Fig. 1c). On the other hand, for analytes with small sizes, such as polystyrene nanoparticles with a diameter of 27.9 nm, the contribution of $2|E_b||E|\cos(\theta)$ becomes significant. Meanwhile, $|E_b|^2$ is static and can be removed in the differential images by subtracting the previous frame from each frame of raw image. In this way, the nanoparticle can be revealed in the differential image at the moment the particle attaches to the sensor surface (Fig. 1d). Note that the background in the differential image in Fig. 1d is not zero due to the shot noise (Supplementary Fig. 4). We measure the polystyrene nanoparticles with different diameters and determine the ESM image intensity by averaging the intensities of all pixels within the Airy disk (Supplementary Fig. 5). The diameter of the Airy disk was estimated to be ~1.07 μm by dividing the incident wavelength of 450 nm with the imaging objective numerical aperture of 0.42. Plotting the image intensity versus nanoparticle diameter in logarithmic scale reveals two regimes, corresponding to large and small nanoparticles (Fig. 1e), where the z-distance dependence of evanescent wave is considered (Supplementary Note 1). In the large nanoparticle regime (diameters > 65 nm), the image intensity follows a power law of $d^{5.6}$, where the exponent is close to six. This is expected because light from the nanoparticles dominates, such that the measured image contrast scales with $|E_s|^2$ according to equation (1). However, in the small nanoparticle regime (<44 nm), the drop in image intensity with decreasing diameter slows down as predicted, because the interference term, $2|E||E|\cos(\theta)$ in equation (1), becomes dominant[25]. The interference effect avoids the rapid drop of image intensity with decreasing analyte diameter, making it possible to detect the single proteins, which are typically smaller than 30 nm.

The main advantage of PSM is that the plasmonic field can provide a significant field enhancement, usually 20–30 times on the gold-water interface[26]. At the same time, the field enhancement is ~5 times for the ESM based on the TIR configuration. Theoretically, PSM can provide ~6 times higher enhancement than ESM at the same excitation wavelength and power with an average period of 50 ms. However, we show that the ESM can overcome this limitation in two ways. First, the cover glass surface has a much lower heating effect than the gold surface (Supplementary Fig. 6), thus allowing the incident intensity up

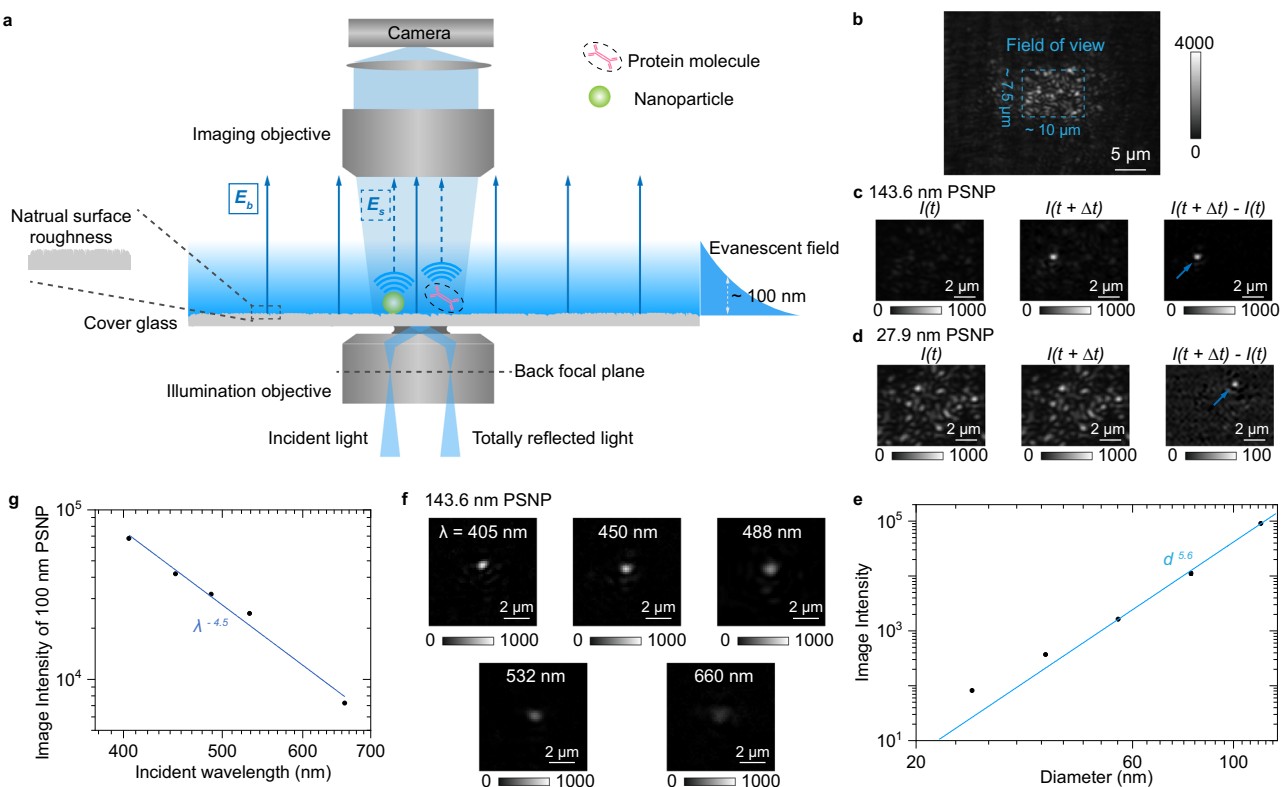

**Fig. 1 Setup and principle of ESM. a** Schematic of the optical setup, where the evanescent field is created by total internal reflection and scattering of the evanescent waves by a particle or protein ($E_s$) and by the glass surface ($E_b$) is collected from the top to form an ESM image. **b** Raw ESM images of a bare cover glass. The blue dashed frame indicates the field of view determined by the illumination area. Incident wavelength: 450 nm. Incident intensity: 60 kW cm$^{-2}$. Camera exposure time: 5 ms. **c, d**, ESM images before and after the binding of a 143.6 nm (**c**) and 27.9 nm (**d**) polystyrene nanoparticle (PSNP), and the corresponding differential images. Individual particles are marked with arrows. Incident wavelength: 450 nm. Incident intensity and camera exposure time are 2 kW cm$^{-2}$ and 0.2 ms for 143.6 nm, and 60 kW cm$^{-2}$ and 1 ms for 27.9 nm, respectively. **e** ESM image intensity versus particle diameter. The image intensity for each diameter was obtained from the mean value of the corresponding histogram in Supplementary Fig. 5, and normalized with an incident intensity of 60 kW cm$^{-2}$ and camera exposure time of 5 ms. **f** ESM images of 143.6 nm PSNP achieved with different incident wavelengths. Incident intensity: 2 kW cm$^{-2}$. Camera exposure time: 0.2 ms. **g** ESM image intensity of the nanoparticle with an effective diameter of 100 nm versus incident wavelength. The image intensity was achieved from the calibration curve as shown in Supplementary Fig. 5 and 8, and normalized with an incident intensity of 60 kW cm$^{-2}$ and camera exposure time of 5 ms. The experiments were repeated three times with similar results for (**b–g**).

to 60 kW cm$^{-2}$, which is ~20 times higher than the commonly used incident intensity for the PSM[17]. Second, the ESM allows employing the short incident wavelength, while the plasmonic field was hard to be excited on the gold-water interface with an incident wavelength shorter than 600 nm (Supplementary Fig. 7). The shorter wavelength can provide a larger scattering cross-section to improve the image contrast (Fig. 1f). Quantitative analysis shows that the ESM image intensity scales with $\lambda^{-4.5}$, where $\lambda$ represents the incident wavelength (Fig. 1g and Supplementary Fig. 8), agreeing with the theoretical prediction of the Rayleigh scattering model (Supplementary Note 2). The incident wavelength of 450 nm was used for the label-free single-molecule imaging in this study because the violet light (405 nm) may damage the surface modification under high intensity (Supplementary Fig. 9). The incident wavelength of 450 nm can provide ~5 times larger scattering cross-section than that of 670 nm, which is commonly used for PSM[17].

**Detection of single proteins.** To demonstrate the capability of ESM for label-free imaging of single proteins, we studied the detection of bovine serum albumin (BSA), mouse immunoglobulin G (IgG), human immunoglobulin A (IgA), human immunoglobulin M (IgM) with ESM (Fig. 2). The measurement was carried out by flowing each protein solution with a 5 nM concentration over the sensor surface while recording the nonspecific

binding of individual proteins on the surface. The surface was modified with N-hydroxysuccinimide (NHS) to increase the binding rate (Methods). Figure 2a shows several frames of binding events of BSA molecules, where the individual proteins are marked with arrows. We tracked and counted individual protein binding events on the differential frames over 5 mins and constructed a protein image intensity histogram (Fig. 2a). The image intensity histogram follows a Gaussian distribution, where the histogram width may result from the protein orientation heterogeneities[27]. Increasing the protein diameter can lead to an increase in ESM image intensities. This is clearly shown by the intensity histograms of BSA, IgG, IgA, and IgM proteins, which have the hydrodynamic diameters of 8.5 ± 2.0 nm, 11.8 ± 1.6 nm, 15.7 ± 2.2 nm, and 21.8 ± 1.9 nm measured by dynamic light scattering, respectively (Fig. 2a–d). The maximum value of the image intensity scale was set to be 1.5 ~ 2 times higher than the maximum intensity of the bright spots created by the proteins on the image for easy reading, and the mean value of the intensities of all pixels included by the bright spots was used to construct the histograms for evaluating the signal intensity more precisely (Supplementary Note 3). To visualize the relationship of protein size with the ESM image intensity, a box plot is provided in Fig. 2e, and Supplementary Video 1 shows the dynamic binding process of these proteins over time in the same grayscale. The mean ESM image intensities of these proteins were obtained by

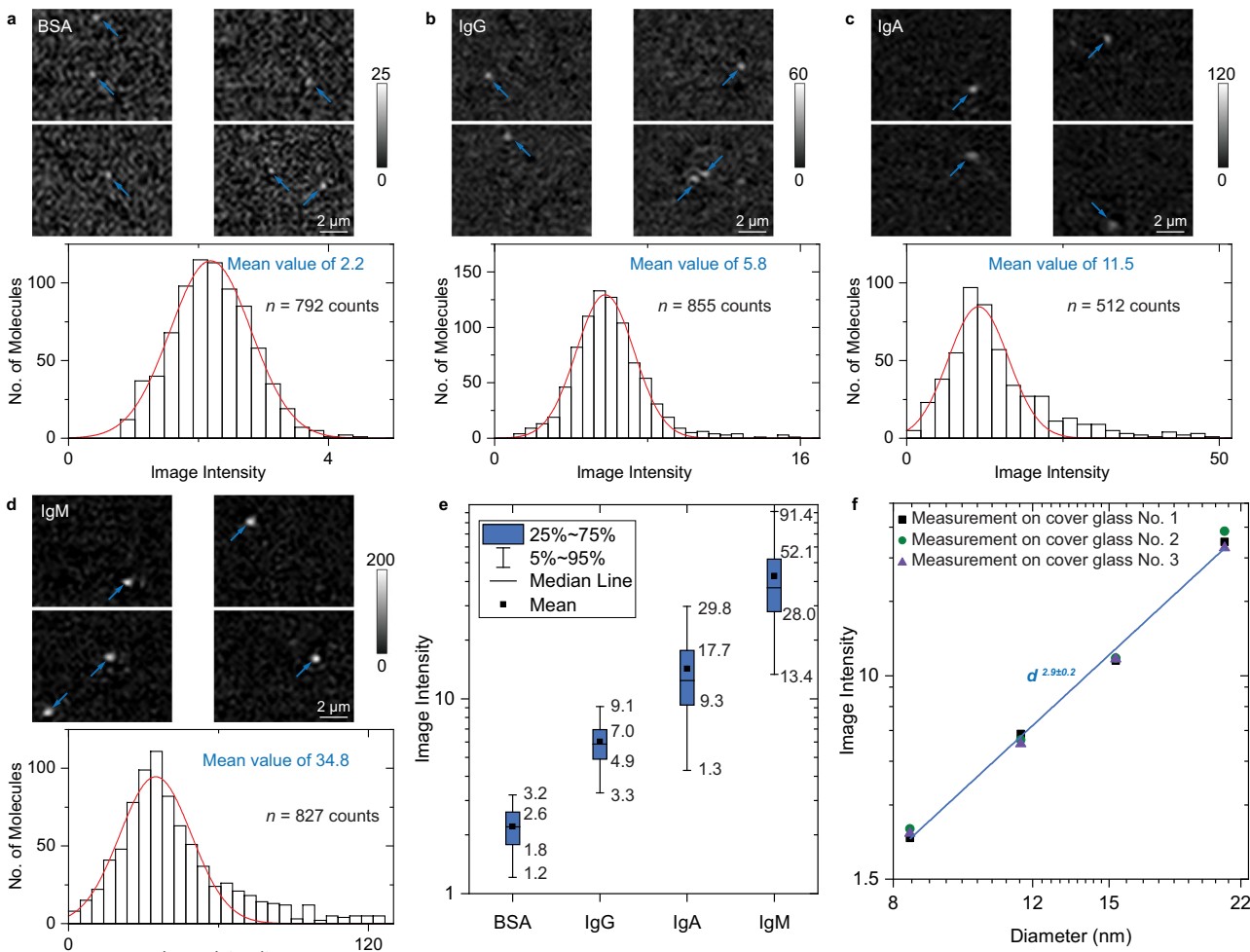

**Fig. 2 Imaging single proteins with ESM. a–d** ESM images and image intensity histograms of BSA (**a**), IgG (**b**), IgA (**c**), and IgM (**d**) proteins, where the solid red curves are Gaussian fittings. Individual proteins are marked with arrows. Incident wavelength: 450 nm. Incident intensity: 60 kW cm$^{-2}$. Camera exposure time: 5 ms. The ESM images of BSA, IgG, and IgA were achieved by subtracting the previous frame from each frame after averaging the raw images with a period of 50 ms. The ESM images of IgM were achieved by subtracting the previous frame from each frame without averaging. The sample sizes and mean values of the histograms are presented in each panel. **e** Box plots of image intensities measured on different proteins from (**a–d**). The sample sizes are presented in a-d, and the counts come from biologically independent molecules. The minima, maxima, and center values are 0.8, 4.4, and 2.2 for BSA, 1.1, 16.0, and 5.9 for IgG, 1.1, 50.0, and 12.4 for IgA, and 1.8, 148.9, and 37.1 for IgM, respectively. The bounds of box and whiskers are presented in the figure. The percentile is defined by mean value, which is 2.2 for BSA, 6.0 for IgG, 14.3 for IgA, and 42.7 for IgM, respectively. **f** ESM image intensity versus protein diameter measured by dynamic light scattering in logarithmic scale, where the z-distance dependence of evanescent wave is considered (Supplementary Note 1). The image intensities for each diameter were obtained from the peak values of the corresponding histograms for measurement on each of the three cover glasses. The histograms obtained from the measurements on different cover glasses can be found in Supplementary Fig. 10. The experiments were repeated three times with similar results for (**a–d**).

fitting the histograms with Gaussian distribution. Reproducible results were obtained for each protein in three different chips (Supplementary Fig. 10). Plotting the image intensity versus protein diameter in logarithmic scale reveals that the ESM image intensity responds to the protein diameter in a cubic power, because the interference term, $2|E||E|\cos(\theta)$ in equation (1), is dominant for single protein imaging (Fig. 2f). We compared the image intensities of proteins with polystyrene nanoparticles (Supplementary Note 2), which fall on a similar calibration curve, further supporting the results of single proteins imaging on ESM.

It is noted that minor peaks show at the position with ~2 times higher intensity than the mean values in several histograms (Fig. 2 and Supplementary Fig. 10). However, with the increase of analyte concentration, the heights of the minor peaks do not scale, and become a long tail at high protein concentrations (Supplementary Fig. 11). We also noted that two proteins may bind to the surface simultaneously at locations close to each other (Fig. 2b). Therefore, the small peaks or tails on the right side of the histogram are likely created by two or more molecules simultaneously falling within the same Airy disk area within a diameter of ~1 μm, which cannot be resolved by the ESM image and counted as a single event. This phenomenon is caused by the finite temporal and spatial resolution of the ESM image, the random binding of molecules to the surface, and possible non-uniform distribution of active carboxylic groups on the sensing surface[28]. We also study the spatial distribution of the ESM image intensity, and the image intensity is slightly higher in the central illumination area than the marginal area due to the Gaussian optical characteristics of the laser beam (Supplementary Fig. 12). This makes it possible to observe smaller proteins, such as protein A molecules, in the central illumination area (Supplementary Fig. 13).

**Quantification of protein–protein interactions**. ESM can image single proteins, making it possible to determine the binding kinetics by digitally counting the association and dissociation of single proteins. As a demonstration, IgA binding to anti-IgA was measured by ESM (Fig. 3a). 20 nM IgA solution was flowed over an anti-IgA coated sensor surface to study the specific association process, and then PBS buffer was flowed over the sensor surface to study the dissociation of IgA from anti-IgA. The association and dissociation processes were tracked by counting the individual IgA molecules in real-time. On exposure to IgA, the specific absorption of single IgA molecules to anti-IgA took place immediately (Fig. 3b and Supplementary Video 2). The negative control experiment was performed by flowing thyroglobulin (Tg, molecular weight of 660 kDa) and IgM over the anti-IgA coated surface. Different from the specific binding of IgA to anti-IgA, the Tg and IgM proteins only transiently show up on the anti-IgA modified surface (Fig. 3b and Supplementary Video 2), and few cumulative binding events were observed (Fig. 3c). The binding kinetic curve can be achieved by plotting the number of cumulative IgA proteins binding to the surface versus time (Fig. 3c). The association ($k_{on}$) and dissociation ($k_{off}$) rate constants are determined to be $1.1 \times 10^5 \, \mathrm{M^{-1} \, s^{-1}}$ and $9.6 \times 10^{-5} \, \mathrm{s^{-1}}$, respectively. The equilibrium dissociation constant ($K_D = k_{off}/k_{on}$) is determined to be 873 pM. These values agree well with the values achieved with the ensemble SPR[17]. The association and dissociation events of individual IgA proteins were counted to construct intensity histograms. The mean intensities by fitting the histograms with Gaussian distributions are consistent with the size of single IgA molecules, confirming the detection of single molecules (Fig. 3d).

The bright spot created by the analyte in the ESM image exhibits a quasi-Airy pattern, thus allowing employing the two-dimensional Gaussian fitting, which has been widely used for single molecule binding sites localization in super-resolution fluorescence microscopy[29]. To correct the intensity variations caused by the uneven background, the mean radial profile was computed from the radial profiles at different angles of one bright spot created by a protein on the ESM image to generate the full-circle point spread function (PSF) (Fig. 3e)[30]. The resulting full-circle PSF was used to cross-correlate with the ESM image sequence to correct the image distortions for improving image quality (Fig. 3f). Then the two-dimensional Gaussian fitting was used for the super-resolution localization with TrackMate[31]. Tracking results on one and multiple IgA molecules show that the super-resolution tracking on the corrected ESM image sequence can provide the precision of ~15 nm, which is close to the theoretical limit (Fig. 3g, h, Supplementary Note 4 for details). Then we can achieve the localizations of cumulative IgA, Tg, and IgM binding events on anti-IgA immobilized on the surface shown in Supplementary Video 2 (Figs. 3i, j). These results clearly show that the specific binding events of IgA to anti-IgA are much more than the nonspecific binding events of others, further demonstrating the capability of ESM for single protein recognition.

**Quantification of DNA conformation changes**. After binding to the antibody, a few proteins may present distinctive behaviors rather than hitting and staying on the surface[17]. For example, Fig. 4a and Supplementary Video 3 show an IgM molecule with repeated hitting behavior after binding on the anti-IgM modified surface, just like 'dancing'. To elucidate the underlying mechanism, we performed the statistical analysis at the single-molecule level. The relation between the ESM image intensity and z-displacement is given according to the exponential distribution of evanescent field intensity (Fig. 4b, and Supplementary Note 5 for

details). Analysis of the region adjacent to the IgM binding sites shows much smaller fluctuations, indicating that this z-axis movement was dominated by the IgM binding (Supplementary Note 5). The statistical distribution shows that the fluctuating z-displacement amplitude increased along with changing the 'binding' condition to the 'dancing' condition (Fig. 4c). The free energy profile, $G(z)$ (the potential of mean force), is related to the probability density of z-displacement amplitude, and the effective spring constant $k_f$ can be obtained by fitting the free energy profiles near equilibrium (Fig. 4d, and Supplementary Note 5 for details)[4,7,32,33]. It can be seen that the $k_f$, reflecting the restoring force of adhesion bond, which is created by molecular binding between IgM and anti-IgM, decreases along with changing the 'binding' condition to the 'dancing' condition, indicating the different adhesion bond properties behind different molecular interaction behaviors.

With the ability to evaluate the restoring force of molecular adhesion bonds, we show that ESM can quantify the conformation changes of DNA from softy single-stranded to rigid double-stranded counterparts after hybridization. This provides a solution for detecting the miRNA molecules, which are challenging to be detected by conventional label-free measurement approaches with practical detection limits for measuring BSA or larger biological macromolecules[10]. We firstly flowed 40 nm gold nanoparticles to the surface with the single-stranded DNA (ssDNA) immobilized to construct the detection system for miRNA measurement (Fig. 4e). The incident wavelength was changed from 450 nm to 532 nm to enhance the scattering via exciting the localized SPR (Supplementary Fig. 14), making the measurement independent of the background created by surface roughness scattering (Supplementary Fig. 15). Then the PBS buffer was flowed over the surface to flush away weakly absorbed nanoparticles, and the z-axis fluctuations of ssDNA-linked nanoparticles were monitored in the PBS flow. Next, the complementary miRNA solution was flowed onto the surface for hybridization with ssDNA to construct a rigid duplex structure. After flushing away the extra miRNA with PBS, the measurement on one rigid duplex structure linked gold nanoparticle in the PBS flow shows that its z-axis fluctuations were obviously smaller than those on ssDNA (Figs. 4e, f). Moreover, the statistical analysis quantitatively shows that the restoring force was increased by ~4 folds due to the DNA conformation changes (Fig. 4g). The clearly decreasing z-axis fluctuations after hybridization could be explained by the increasing restoring force after rigid duplex structure formation (Supplementary Note 6)[32]. To make our analysis more convincing, more gold nanoparticles linked by ssDNA immobilized on another cover glass were tracked and analyzed. Both spatial distribution and statistical analysis show that their $k_f$ increases after hybridization with miRNA (Fig. 4h and 4i). The measurement results can be repeated on different cover glasses (Supplementary Fig. 16). The background analysis also indicates that the nanoparticle z-displacement variations should be induced by the thermal fluctuations of adhesion bonds, rather than the molecular diffusion or experimental conditions (Supplementary Notes 5 and 6). These results demonstrate that ESM can perform miRNA detection by monitoring the change in z-axis fluctuations of nanoparticles caused by hybridization-related conformation change.

## Discussion
We developed ESM for single-molecule imaging based on TIR, which is an ancient method but still being developed for new applications[19,20,34,35]. In contrast with far-field interference imaging methods[11,21,36–41], ESM has image contrast arising from the

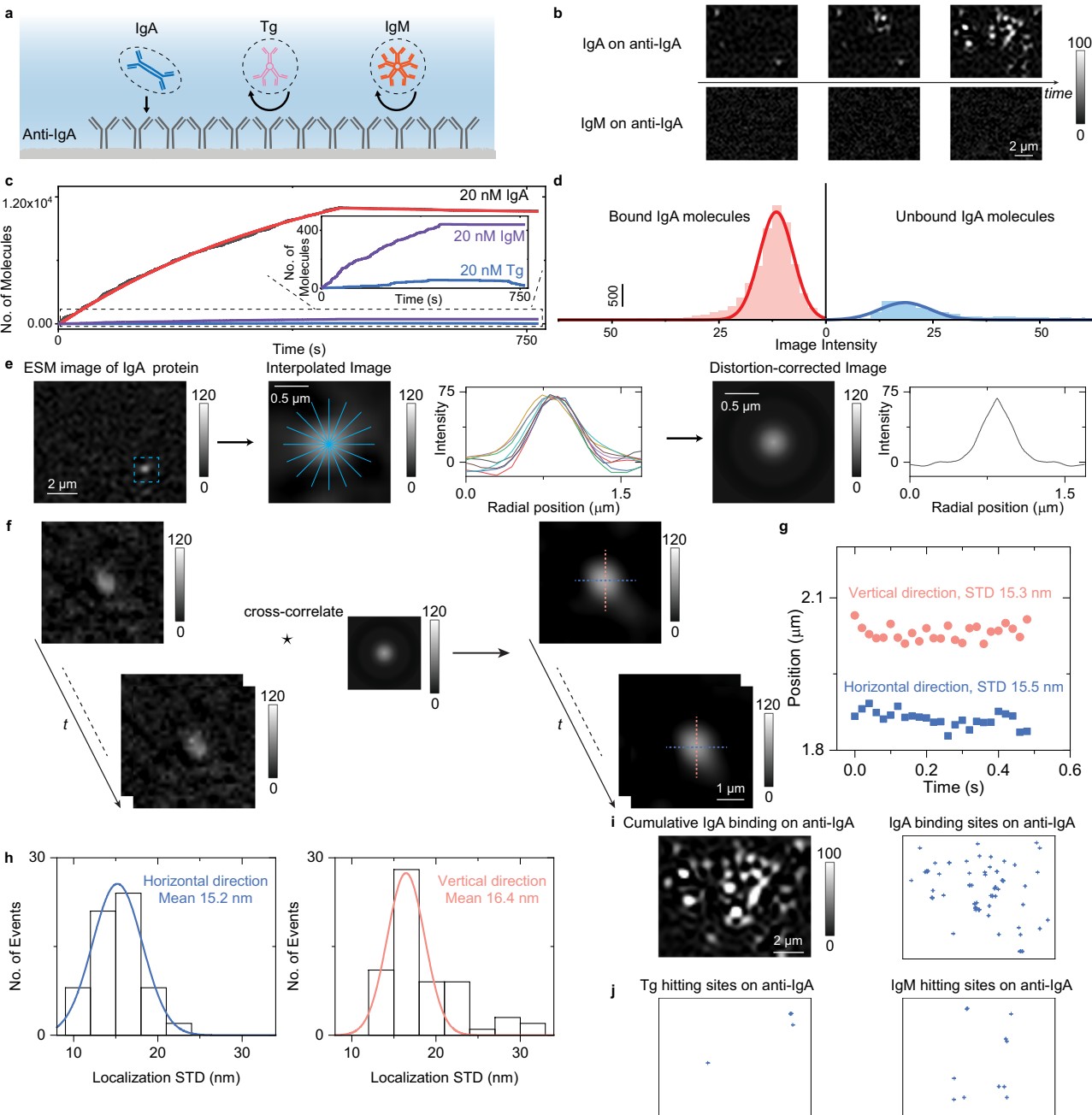

**Fig. 3 ESM identification of single proteins using antibodies. a** Schematic showing the behaviors of single IgA, Tg, and IgM proteins on the anti-IgA modified surface. **b** ESM images showing binding of IgA to anti-IgA immobilized on the surface, and negative control experiment, exposing of IgM to anti-IgA surface. **c** Kinetics of 20 nM IgA binding to anti-IgA determined by digital counting of the binding/unbinding of single IgA molecules (black curve and solid red fitted curve), and negative control experiments exposing of Tg and IgM to anti-IgA surface. Inset are zoom-in view of the control experiment results. **d** Image intensity histogram obtained from association, and dissociation processes of 20 nM IgA injected onto anti-IgA modified surface, where the solid lines are Gaussian fitting. **e** Extracting mean radial profiles from one differential frame where one protein has been recognized to construct full-circle point spread function (PSF). **f** Correcting the ESM image distortions by cross-correlating the ESM images with the full-circle PSF. **g** Tracking the position of one protein binding on the sensor surface with an effective frame rate of 50 fps after average and duration of 0.5 s using TrackMate. STD represents the standard deviation. **h** Localization standard deviations (STD) histograms achieved by tracking different molecules. **i** Super-resolution image, showing the localized positions of the cumulative IgA binding events on anti-IgA immobilized on the surface. **j** Super-resolution image, showing the localized positions of the Tg and IgM hitting events on anti-IgA immobilized on the anti-IgA immobilized on the surface. Incident wavelength: 450 nm. Incident light intensity and camera exposure time are 20 kW cm$^{-2}$ and 10 ms. The image intensity was normalized with an incident light intensity of 60 kW cm$^{-2}$ and a camera exposure time of 5 ms. The experiments were repeated three times with similar results for b and i.

interference of evanescent light scattered by an analyte and the surface roughness, a principle has been proposed for PSM[17]. As a result, both ESM and PSM can achieve comparable signal-to-noise ratios with either lower incident light power or a wider field of view than conventional nonevanescent approaches. Compared to PSM, ESM shows two advantages. First, the choice of incident wavelength is more flexible for the ESM owing to the TIR configuration, while only the red or longer incident light can excite SPR at the

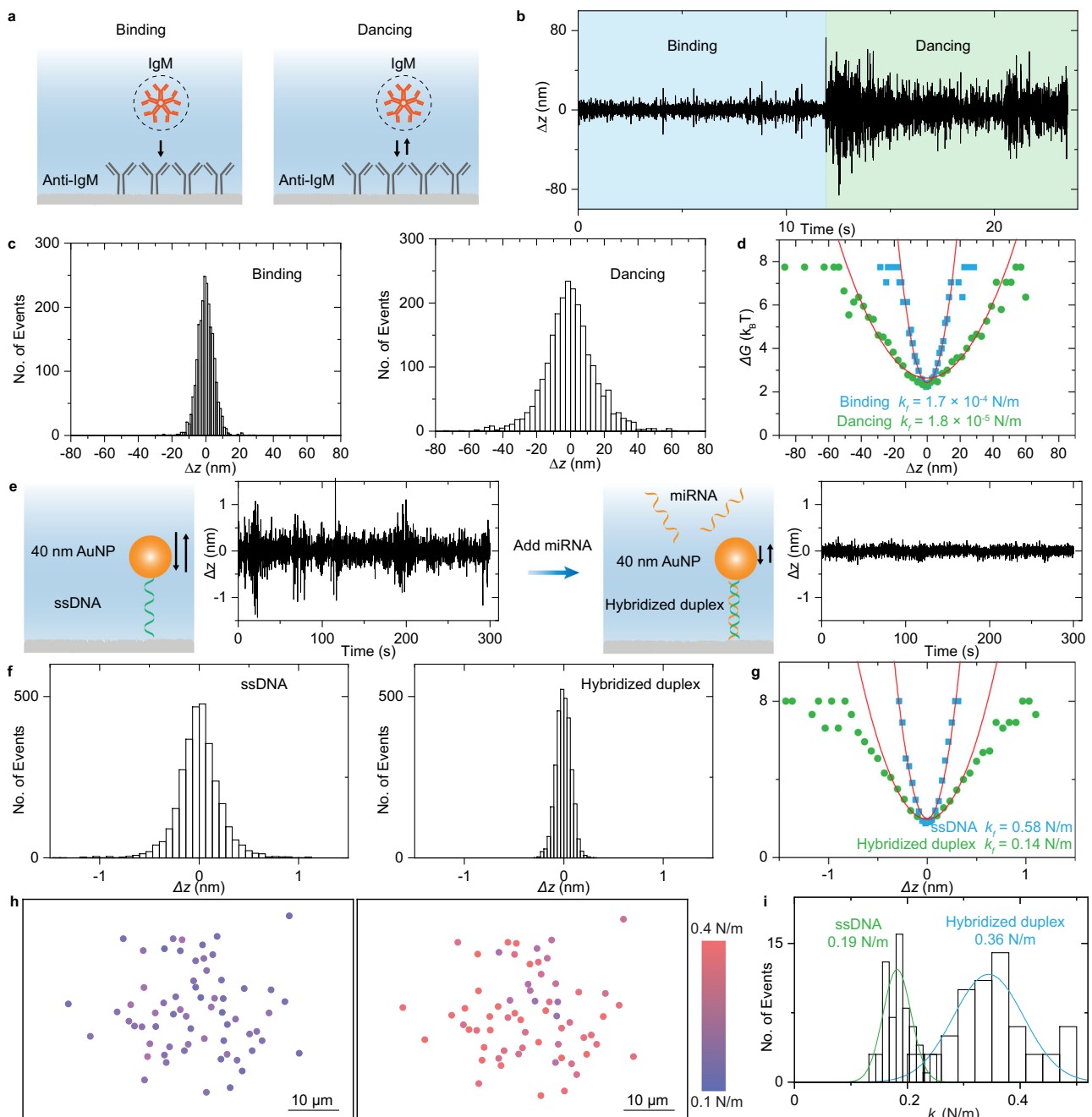

**Fig. 4 Quantification of DNA conformation changes. a** Two types of protein interaction modes on antibody-modified surface: hit and stay (binding), and bind and unbind rapidly (dancing). **b** Time trace of z-displacement of one IgM molecule on the anti-IgM modified surface under two protein interaction modes. **c** Probability density determined from the statistical distribution of z-displacement amplitude. **d** Free energy profile achieved from the probability density and fitted results with a polynomial function. The effective spring constant obtained from fitting was also presented. For Fig. 4a–d, incident wavelength is 450 nm, and incident light intensity and camera exposure time are 60 kW cm$^{-2}$ and 5 ms. **e** Schematic showing the structure of ssDNA anchored gold nanoparticle (AuNP) on the glass surface before and after hybridization with miRNA. Time trace of z-displacement of one nanoparticle linked by the soft ssDNA and rigid hybridized duplex structures was also presented. **f** Probability density determined from the statistical distribution of z-displacement amplitude for the gold nanoparticle shown in Fig. 4e. **g** Free energy profile achieved from the probability density and fitted results with a polynomial function. The effective spring constant obtained from the fitting was also presented. **h** Effective spring constant colormap for 64 gold nanoparticles linked by ssDNA molecules on the surface before and after hybridization with miRNA. **i** Effective spring constant statistical distribution for 64 gold nanoparticles linked by ssDNA molecules on the surface before and after hybridization with miRNA, where the solid curves are Gaussian fitting. For Fig. 4e–i, incident wavelength is 532 nm, and incident light intensity and camera exposure time are 2 kW cm$^{-2}$ and 0.2 ms.

gold-water interface on PSM. This makes it possible to employ 450 nm as incident light for ESM, achieving ~5 times larger scattering cross-section than PSM using red incident light. This advantage allows TIR to achieve comparable evanescent wave

scattering intensity with PSM, which is usually ~6 times larger than the former under the same incident intensity. Second, ESM is constructed on the cover glass, which absorbs fewer lights than gold film, thus allowing the incident light intensity of up to 60 kW cm$^{-2}$

without a significant heating effect. This incident intensity is ~20 times stronger than the upper-limit incident intensity of PSM. While PSM can only image large proteins as small as IgG (150 kDa) limited by the heating effect, ESM can reliably image medium-sized single proteins as small as the BSA (66 kDa), which is usually considered as the practical measurement limit for label-free single-molecule detection technologies[10]. Therefore, ESM can measure a much wider range of proteins than PSM.

In addition, one approach was presented for the lateral super-resolution localization of protein binding sites in the ESM image. The intensity fluctuations resulting from the non-uniform background were corrected, and the binding sites of different proteins on the antibody-modified surface were localized. As a result, the super-resolution image shows the protein binding positions with high precision, which clearly reveals the interaction frequency difference between specific and nonspecific binding events. This super-resolution localization approach could be applied to study spatial variation of surface binding events with high resolution.

We demonstrated that ESM could also provide a label-free detection scheme for small molecules, such as miRNA. The axial diffusion of gold nanoparticles linked to the surface through ssDNA can be tracked by ESM with at least 0.04 nm resolution. This is due to the exponential distribution of evanescent field intensity, and the conformation changes resulting from miRNA hybridization with ssDNA can be measured by analyzing the z-axis thermal fluctuations of ssDNA-linked nanoparticles. Although the axial tracking resolution of 0.02 nm has been reported on the SPR system[42], the electron transfer between gold nanoparticles and the gold film will lead to the nonlinear variations of SPR scattering signals[43], making it challenging to be realized with the conventional optical system and image processing approaches. This chain molecular linking nanoparticle tracking scheme provides a solution for the small molecule detection in a regular buffer.

We have demonstrated the evanescent single-molecule imaging with TIR configuration. The light scattered by the natural roughness of cover glass is used as a reference for interfering with light scattered by proteins, providing the quantitative size imaging of single proteins. We show that this approach can analyze the heterogeneities of single-molecule binding behaviors and permit the analysis of DNA conformation changes by tracking the axial fluctuations of linking nanoparticles. In addition, the cover glass has a lower heating effect, no plasmonic quenching effect, and good optical clearance so that the ESM can be easily integrated with fluorescence imaging for multiplexed detection in future applications (Supplementary Note 7). Thus, we anticipate that the ESM can become an essential tool for analyzing single molecules and biological complexes' behaviors, especially when combined with volumetric fluorescence imaging to understand cell activities systematically.

## Methods

**Materials**. The No.1 cover glasses (22 × 22 mm, Catalog No. 48366-067) and isopropyl alcohol (IPA, Catalog No. BDH2032-1GLP) were purchased from VWR (Radnor, PA, US). Polystyrene nanoparticles were purchased from Bangs Laboratories (Fishers, Indiana, US). Spherical gold nanoparticles (Catalog No. A11) were purchased from Nanopartz company (Loveland, CO, US). 0.1% poly-l-lysine (Catalog No. P8920), hydrogen peroxide (H₂O₂, 30%, Catalog No. H1009), (3-Aminopropyl)triethoxysilane (APTES, Catalog No. 440140), succinic anhydride (Catalog No. 239690), bovine serum albumin (BSA, Catalog No. A7638), human thyroglobulin (Tg, Catalog No. T6830), sodium hydroxide (NaOH, Catalog No. S5881), monoclonal anti-dinitrophenyl antibody (Catalog No. D8406) and Tris(2-carboxyethyl)phosphine (TCEP, Catalog No. 646547) were purchased from Sigma-Aldrich (St. Louis, MO, US). Ammonium hydroxide (NH₃·H₂O, 28.0–30.0%, Catalog No. C5103500-2.5D) was purchased from Mallinckrodt Reagents (Belmont, NC, US). 1-ethyl-3-(3-dimethylaminopropyl)carbodiimide hydrochloride (EDC, Catalog No. 22980) and Sulfo-NHS (N-hydroxysulfosuccinimide, Catalog No. 24510) were purchased from Thermo Scientific (Waltham, MA, US). Phosphate-Buffered Saline (PBS, Catalog No. 21-040-CV) was purchased from Corning (Corning, NY, US) and filtered with 0.22-μm filters (Millex-GS, Catalog No. SLGSM33SS) from Sigma-Aldrich (St. Louis, MO, US). Tris-EDTA (TE,

Catalog No. BP2476) was purchased from Fisher Scientific (Hampton, NH, US). Native protein A (Catalog No. ab7399) was purchased from Abcam (Cambridge, UK). Anti-IgA (IgG, Catalog No. STAR141) was purchased from BIO-RAD (Hercules, CA, US). Human colostrum IgA (Catalog No. 16-13-090701) and Human IgM (Catalog No. 16-16-090713) were purchased from Athens Research and Technology (Athens, GA, US). Deionized water with resistivity of 18.2 MΩ cm⁻¹ was filtrated with a 0.22-μm filter and used in all experiments. The monoclonal anti-dinitrophenyl antibody was diluted by 1000 times to 1 μg/mL for the experiments. The anti-IgA was diluted by 300 times to 20 nM for the experiments. The ssDNA and miRNA were purchased from Bio-Synthesis Inc. The ssDNA was modified with a disulfide bond at 5′ for nanoparticle conjugation and NHS ester at 3′ for linking to the glass substrate. The ssDNA and miRNA sequences are as follows:

5′ [NHS][Amino C6 linker]-AAC CCC TAT CAC GAT TAG CAT TAA TTT-(CH2)3-S-S-(CH2)3 3′

miR-155 5′-UUA AUG CUA AUC GUG AUA GGG GUU-3′

**Experimental setup**. An 80-mW laser diode (PL450B, Thorlabs, Newton, NJ, US) is used as the light source to provide the incident wavelength of 450 nm. The laser diode is fixed at a temperature-controlled mount (LDM38, Thorlabs), which is driven by a benchtop diode current controller (LDC205C, Thorlabs) and a temperature controller (TED200C, Thorlabs). Light from the laser diode is conditioned by an achromatic doublet lens group, and then focused on the back focal plane of a × 60 objective (Olympus APO N 60x Oil TIRF, NA 1.49) by a tube lens with a focal length of 300 mm. The incident angle was adjusted by a manual translation stage to reach total internal reflection condition at 65° (XR25P-K2, Thorlabs). Reflection light is also collected by a camera (MQ013MG-ON, XIMEA) for helping to find the objective focus position. Scattered light from the protein and glass surface is collected by a × 50 objective (NA, 0.42) to form an ESM image by a second camera (MQ003MG-CM, XIMEA). Coherent OBIS FP 405 LX, OBIS FP 488LS, OBIS FP 532LS, and OBIS FP 660 LX lasers were used as the light source to provide the incident light with central wavelength at 405, 488, 532, and 660 nm. A detailed schematic representation of the optics can be found in Supplementary Fig. 1. A thin cover glass constructed flow cell with ~50 μm channel height was employed for sample delivery as previously reported[16,17], and the flow rate is controlled to achieve laminar flow with a gravity-based pumping system (Supplementary Note 6).

**Surface functionalization**. For polystyrene nanoparticle detection, the cover glass surface was incubated with 0.1% poly-l-lysine for 1 h. Then the surface becomes positively charged to achieve a high binding rate. For measuring the nonspecific binding of single proteins, the cover glass was modified with active carboxyl groups using the following steps. (1) The cover glasses were cleaned in the boiling solution mixing the NH₃·H₂O, H₂O₂, and water with a volume ratio of 1:1:5 for 1 h to obtain hydroxylated glass surfaces, where dropping water became a layer. (2) The cover glasses and container were washed twice with water, and then ultrasonically cleaned two times with water, and blew dry with nitrogen. (3) The hydroxylated cover glasses were incubated in boiling 1% APTES diluted with IPA for 3 h to functionalize the surface with the primary amine group. After processing, both solution and cover glass should be clear if drying in the second step is performed correctly. (4) Cover glass and container were washed twice with IPA, ultrasonically cleaned twice with IPA, and blew dry with nitrogen. (5) Incubate the amino group modified cover glasses in 10 g L⁻¹ succinic anhydride in water for 1 h to obtain carboxylic group functionalized cover glass chips. The pH of succinic anhydride solution is adjusted to 7.5–8 with 1 M NaOH solution. The cover glass and container were washed twice with water, ultrasonically cleaned twice with water, and then stored in the water prior to use. In the experiment, the surface was incubated in 40 g L⁻¹ EDC mixed with 11 g L⁻¹ Sulfo-NHS for 15 min to activate the carboxyl groups for connecting proteins. The EDC/NHS solution was filtered by a 0.22-μm filter before use. For specific binding kinetic analysis, the activated carboxylic group modified cover glasses were rinsed with PBS buffer, and then 20 nM anti-IgA or anti-IgM in PBS buffer was applied to the surface and incubated for 1 h to allow immobilization. Next, the surface was incubated in 1 mg ml⁻¹ BSA for 10 min to block any remaining activated carboxylic group to minimize nonspecific binding. Finally, the protein solution was flowed onto the surface for specific binding measurement.

For the nanoscale tracking of ssDNA-linked nanoparticles, the ssDNA was linked to the glass surface through amine-NHS reaction by flowing the chamber with 250 μL of 7.5 μM ssDNA solution over 1 h and followed by washing with 2 mL of PBS over 10 min. Then, the ssDNA was reduced to expose the –SH bond for connecting the gold nanoparticles by reductive cleavage of the disulfide bond: 3 ml of cleavage buffer comprising 50 mM of the reducing agent TCEP in PBS was introduced to the chamber for 15 min. After reduction, the chamber was flushed with 2 mL of PBS for 10 min to exclude any residual cleavage buffer. Finally, the miRNA and ssDNA hybridization were conducted by flowing 500 μL of 6 μM miRNA in TE buffer across the chamber for 11 min. The sample was then washed by flowing 2 mL of TE buffer over 9 min. All buffers were freshly prepared before each experiment using nuclease-free water.

**Data processing**. The raw image sequence captured at a high frame rate (~100–1,000 fps) was converted to an averaged-image sequence, by averaging

images over every 20 ms using previously reported MATLAB program[17] or the real-time averaging function of the camera recording software (XIMEA CamTool), to suppress shot noise. To remove the background, a differential image sequence was obtained by subtracting the previous frame from the present frame of the averaged-image sequence using the MATLAB program shown in Supplementary Note 8. The TrackMate plugin in ImageJ was employed to find and count particles or molecules[31]. The ESM intensity of a particle or molecule was determined by average the powers of all pixels within the Airy disk (Supplementary Note 2). Origin 2019 was used to create data plots and histograms. Scrubber v.2.0a was used to determine the association and dissociation rate constants by fitting the curves in Fig. 3c with the first-order binding kinetics model.

**Reporting summary**. Further information on research design is available in the Nature Research Reporting Summary linked to this article.

## Data availability
Source data of Figs. 1–4 are provided. The raw data that support the findings of this study are available from the corresponding author upon request.

## Code availability
MATLAB code for differential processing is provided in Supplementary Note 8. MATLAB codes for cross-correlation processing are deposited on GitHub and are available at https://doi.org/10.5281/zenodo.6471541.

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

## Acknowledgements

We thank the financial support from the National Institute of General Medical Sciences of the National Institutes of Health grant R01GM107165 (S.W.). The content is solely the responsibility of the authors and does not necessarily represent the official views of the sponsors.

## Author contributions

P.Z. performed the optical experiments and data analysis. L.Z. contributed to the surface chemistry and DNA experiments. R.W. and X.Z. contributed to the cell culture and fluorescence imaging. R.W., L.Z and J.J contributed to heating effect measurements. Z.W. performed the atomic force microscopy measurement. S.W. conceived and supervised

the project. P.Z., L.Z., R.W. and S.W. wrote the manuscript. All authors reviewed and commented on the manuscript.

## Competing interests

A US provisional patent application (63/249,388) has been filed by Arizona Board of Regents on behalf of Arizona State University for single-molecule imaging based on an early draft of this article. Inventors are S.W. and P.Z. The remaining authors declare no competing interests.
