## [Peer Review File · Nature Communications]

Reviewers' Comments:

Reviewer #1:

Remarks to the Author:

The manuscript by Zhang et al describes the use of evanescent scattering for imaging single molecules, their motion and their kinetics. The work is an extension of recently published work (Nat. Methods 17, 1010 2020) using an identical experimental approach, except that the sample used does not involve a gold coated glass cover slide. Nevertheless, the observation is a significant advance and in principle worthy of publication in nature communications. The major problem here is that many conclusions made in the paper as written are not supported by the data because insufficient data and statistical analysis is given. All too often it feels like the authors performed an assay to demonstrate some property and then show a small data set that fits that hypothesis. It is not, however, at all clear to which degree the observed quantities are (a) real, and (b) reproducible. In conclusion, the observation of single molecules in this fashion is exciting, but the measurements performed and conclusions drawn are too far apart for publication in nature communications - or any other journal - at this stage. I thus strongly recommend the authors go back and perform repeats and detailed statistical analysis of their results.

1. Given the extreme similarity between this work and the recent nature methods publication (essentially identical experimental setup), it would seem appropriate to provide a more explicit comparison between the methods. The authors need to acknowledge that they (and others) have suggested for decades that surface plasmons are essential to achieving high sensitivity, yet here they suggest that they can achieve the same sensitivity without a gold film supporting surface plasmons. I had to dig significantly in SI of this and their other paper to realise that indeed the incident power density is 20 times higher here than in the other publication. This discussion and explicit comparison needs to be much clearer and included in the main manuscript, ideally on page 9 where some comparisons are made, but much is left to be desired. The authors have to be explicit and upfront about the implications of the gold film 'apparently no longer being needed' after it was apparently crucial for two decades.

2. I struggle very much with the arguments made in SI Fig N1.3 - which are nevertheless essential to understand the source of contrast. If I read the figure correctly, the authors suggest that there is a difference of a factor of 2 between the scattering from IgA and IgM, which matches a d^3 scaling with diameter. This would suggest that the primary factor for determining contrast is diameter. Yet in Figure 2f they suggest that the signal scales perfectly with molecular weight. For the same two proteins, they report perfect correlation, including a difference in image intensity of a factor of ~ 1.5 for proteins with a mass difference of a factor of almost three. These data contradict each other: what does the signal scale with, mass or diameter? And if it scales with mass, how does the 26 nm PSNP fit in here? I would expect the mass of a 26 nm PSNP to be much, much larger than of IGM, yet the image intensity is only a factor 2 larger? Also, proteins can have very similar mass and very different diameter and vice versa - so what matters and why?

3. Fig2 is clearly very important, but also has a number of quite serious issues that must be addressed.

(A) The correlation in Fig. 2f appears perfect. Is this really true? How many measurements were used to obtain the data point and error bar? 1, 2, 3, 5? It appears 1. But in SI Fig N3.2 the authors also show that there is very large variation with cover glass roughness (50% difference in contrast between different measurements). So how did they manage to get such perfect correlation. This very much suggests that the authors conveniently picked histograms. The authors must show several repeats for each protein so that we can understand the true variations in the measurement. The results of each of these repeats should be incorporated in Fig 2f and ideally shown as a box plot including all the data points. In addition, if the relationship is indeed linear as suggested in the figure, I do not understand why the axes chosen to be logarithmic - the data would be much clearer with linear axes given that we are covering only about 1 order of magnitude in both dimensions.

(B) Fig 2f suggests that they could resolve proteins such as BSA, IgG and IgA given the shown error bars. Yet looking at Figs 2b, c and d, that appears clearly impossible. The error bars do not seem representative. This needs to be checked.

(C) The authors suggest that they are detecting dimers of proteins in all histograms, where they

seem to simply add a Gaussian where a dimer would be expected. For this measurement to be meaningful, they need to (a) measure at different concentrations, which should change the amount of dimer; and (b) would need to show several repeats of the same protein to be able to judge to which degree the observed shoulders are statistically significant.

4. There is insufficient clarity as to the meaning of the results shown in Fig. 3d-f. The authors suggest that they observe 'dancing' molecules, and then go on to characterise the associated kinetics and thereby the affinities. It is not clear how the authors decided where the border between bound and unbound was (it appears about halfway between 0 and the maximum observed intensity) but why? Moving this line up or down would majorly affect the measured kinetics and thus K_d . The authors further state that what they are observing is likely due to surface bound species/a saturated surface. Ultimately, this is an artefact - and I am unconvinced of the value of these measurements for this paper. If they want to demonstrate that they can measure on and off rates in a meaningful way, it appears that they should choose systems with different affinities/measured rates and demonstrate that their measurement agrees, not in the way that it is done here. In addition, are the authors sure that dancing molecules are not simply molecules where the background is fluctuating, and thus the signal is fluctuating?

5. I am surprised to see the results in Fig. 4: The results shown in Fig. 4b suggest that the authors can achieve sub-nm localisation precision, which is what would be required for the dots to be meaningful, and thus the extraction of diffusion coefficients to be meaningful. I do not understand how this could possibly be the case given the highly non-uniform PSF of the method. In addition, I am very surprised by the very low diffusion coefficient observed - it would seem that the motion of such a small molecule on such a small tether would completely average out on the timescale of the measurement. As previously, how reproducible are these results? It appears that the data shown is from a single assay. How do the distributions in c,d and e vary from assay to assay, with different surfaces, on different days? One could completely explain Fig. 2b by a slight change in localisation precision/fitting quality/focus drift after addition of mRNA. Have the authors done controls where they flowed in buffer not containing miRNA to see if that affects the measurement? Are the differences before and after addition of miRNA reproducible for multiple experiments? In addition, I could not find any information on how the tracking was actually done - which seems essential. I would be surprised if feeding the observed PSFs into track mate could provide any meaningful results whatsoever.

6. I do not understand, and the authors would need to clarify, how it is possible that supposedly the signal intensity is independent of local image background (SI Fig 4), but dependent on overall image intensity (SI Fig N3.2)? The author clearly state and suggest that their measurement is interferometric, combining scattered and background light. How then, can they generate the same signal when the protein lands in a dark vs a bright region of the glass? That seems completely impossible.

7. I struggle to understand some of the observations in the supplementary videos. It appears that the authors use a relative large illumination area (although I could not see it defined). Why is it that Protein A only binds in a 2x2 micron area, BSA in a 4x4 micron area, and IgA apparently everywhere? This is very confusing indeed and needs to be clarified/discussed.

8. The colour scale chosen for the images in the main manuscript (blue to orange) is very poor. It makes it near impossible to see/judge the image quality. This should be optimised.

Reviewer #2:

Remarks to the Author:

P. Zhang et. al. developed the evanescent scattering microscopy (ESM) based on total internal reflection (TIR) concepts to perform single-molecule imaging on plain glass surfaces. The authors have demonstrated their technique by quantifying the mass of single proteins, studying the ligand interactions, and measuring the diffusion properties of single molecules. I do not recommend publishing this manuscript mainly because of the following two reasons,

1) These concepts and experiments carried out in this manuscript are similar to their previous

paper [Nature Methods 17, 1010–1017 (2020), Reference 21 in the manuscript] where the authors developed plasmonic scattering microscopy by utilizing surface plasmon resonance (SPR) concept. The key difference in this manuscript is that instead of using SPR, TIR was used with the glass surface. The optical setup, image processing, and even the experiments are very similar to the previous paper. I agree with the authors that using glass surfaces will inhibit plasmonic quenching and heating, however, I don't believe this as a novel advancement to report in Nature Communications. In addition, no new scientific insights have been obtained on the single-molecule studies as compared to their previous paper as well as to the other single-molecule imaging reports.

2) The references were not properly cited, it looks like authors have given more importance to the journal name/IF/publishing group more than the actual science itself. First, since the manuscript is submitted to Nature Communications, 46% of the cited references are from Nature journals. Second, many of these references look like they were forcibly cited, I would like to give two examples – a) Ref 1 has nothing to do with single-molecule detection or biological interaction, and b) In page 4, the authors mentioned that they have used TrackMate plugin for analysis and cited Ref 34 and 35, in which 35 is relevant whereas 34 does not discuss the specific plugin. If authors want to cite the ImageJ, they can cite the website directly.

Reviewer #3:

Remarks to the Author:

Overall, I find the paper very interesting and worthy of publication. However, a careful revision is needed for clarifying the ideas and results presented.

Review

Evanescent Scattering Imaging of Single Molecule Binding Kinetics and Nanoscale Motion by Pengfei Zhang, Lei Zhou, Rui Wang¹, Xinyu Zhou¹, Zijian Wan and Shaopeng Wang.

I find the paper interesting and worthy of publication. The technique proposed and demonstrated offers important advantages and should be of interest to the single molecule sensing community.

However, the writing of the paper needs improvements. I

Major points:

Please respond and make amendments to the paper accordingly. (Text reproduced from the manuscript is put in blue color.)

1. The main innovation in the experimental technique is using the interference of light scattered by natural surface roughness of the glass slide and light scattered by molecules immersed in the evanescent fields near a glass interface when illuminated from the glass side at an angle of incidence larger than the critical angle. However, little is said about the roughness parameters of the interface, other than presenting an atomic force line scan (supplementary Fig. 2) and that the SNR is independent of the roughness parameters. A discussion about the requirements of the roughness is needed in the main text. Is it just a happy coincidence that bare glass-slides have the appropriate roughness? Can it be intentionally improved? On what does the phase difference, θ , depends? Is the same for all pixels in the images? How does it depend on the position of the molecule, on the roughness and the viewing angle?
2. Figure 1 in the main text is essential to understand the paper and needs more detail. May be supplementary Fig. 1 should be part (b) of figure 1 in the main text. In the current Fig. 1 it appears that a well collimated laser beam is incident at a well-defined angle of incidence. However, in supplementary Fig. 1 one can see that light is focused on the interface, and light is incident at a range of angles of incidence around 65 degrees. This means that there is evanescent light with different penetration lengths illuminating the other side of the interface.
3. Related to the latter point, it appears to me that the angles of incidence (since it is not a single angle of incidence due to focusing) is a crucial parameter that needs to be set with care. The penetration length depth depends on the angle of incidence. Some discussion about this is given in the supplementary notes. However, something must be said in the main text.
4. It is not clear to me what does the intensity histograms in Fig. 2 mean and how they must be interpreted. What determines the “total intensity” in an image at a protein binding site? Why some molecules produce one total intensity and other molecules produce other total intensity?

Minor, but mandatory, points:

The following sentences are not clear and need revision. (Text reproduced from the manuscript is put in blue color)

1. The words “diffraction limit” are used through the paper. A clear definition of what it is meant by this is needed. What is its value in the current experimental setup?
2. Top of page 2: “Single molecule detection is required to analyze the heterogeneous and stochastic processes for deep understanding of mesoscopic scale biological interactions at the level of detail.” Specifically, what it is meant by “heterogenous process” and by “scale biological interactions at the level of detail” Not clear.
3. Near the bottom of page 2: “..wide field imaging applications” What are wide field applications?
4. Middle of page 3: “In addition, we show that ESM can provide high resolution images by eliminating the parabolic tails, which is a common issue for the traditional evanescent imaging system” What are “parabolic tails” please clarify.
5. Figure caption of supplementary Fig. 3 is not clear to me: “The raw image sequence of a bare cover glass was recorded at 200 frames/s for 30 s. Then, differential image sequences with different image average times were obtained. Next, total intensity of all pixels in a selected airy disk sized spot (~6 pixels diameter circled area) for all images in the differential image sequences were calculated. Standard deviation of the total intensity in each 30 s differential image sequences are calculated and plotted as black dots. The red line is expected shot noise value based on the total photons collected in the spot. Therefore, for the averaging time below 50 ms, shot noise is dominant. Incident light intensity, 60 kW cm⁻². Exposure time, 5 ms.” What is it meant by: “raw image sequence”, “Differential image sequence”, “total intensity” and “Airy disk size spot”
Looking at the main text it says “Then, differential images were achieved by subtracting a previous frame from each frame to remove $|Eb/2$.” Shouldn't we get zero?
6. In caption of supplementary Fig. 4 the sentence: “The small second peak (blue line) is attributed to formation of dimers or two particles binding to the nearby surface simultaneously with distance smaller than the diffraction limit.” is confusing. Clarify “distance smaller than the diffraction limit”. Also, the word “corrcoef” near the end of the caption must be corrected.
7. In the appendix of supplementary note 1: Provide a reference for the equation. Explain how this equation is obtained. Define E_{sp} . How are effective diameters calculated? Some of these ideas must be mentioned in the main text of the paper.
8. In supplementary Note 4: How is the equation provided obtained? What is it meant by “Tracking precision”. σ is the standard deviation of what? How is it possible to obtain particles' displacements in the order of 1 nm with an image of about 6 microns wide?
9. Supplementary Note 6: Please cite references for the equations used.

Reviewers' Comments:

Reviewer #1:

Remarks to the Author:

The manuscript by Zhang et al describes the use of evanescent scattering for imaging single molecules, their motion and their kinetics. The work is an extension of recently published work (Nat. Methods 17, 1010 2020) using an identical experimental approach, except that the sample used does not involve a gold coated glass cover slide. Nevertheless, the observation is a significant advance and in principle worthy of publication in nature communications. The major problem here is that many conclusions made in the paper as written are not supported by the data because insufficient data and statistical analysis is given. All too often it feels like the authors performed an assay to demonstrate some property and then show a small data set that fits that hypothesis. It is not, however, at all clear to which degree the observed quantities are (a) real, and (b) reproducible. In conclusion, the observation of single molecules in this fashion is exciting, but the measurements performed, and conclusions drawn are too far apart for publication in nature communications - or any other journal - at this stage. I thus strongly recommend the authors go back and perform repeats and detailed statistical analysis of their results.

Response:

Thank you so much for the positive feedbacks on the novelty of ESM and suggestion for reproducibility experiments. We have added substantial new experimental results and discussions to confirm that ESM can reproducibly detect single molecule/particles as detailed below each comment.

Comments

1. Given the extreme similarity between this work and the recent nature methods publication (essentially identical experimental setup), it would seem appropriate to provide a more explicit comparison between the methods. The authors need to acknowledge that they (and others) have suggested for decades that surface plasmons are essential to achieving high sensitivity, yet here they suggest that they can achieve the same sensitivity without a gold film supporting surface plasmons. I had to dig significantly in SI of this and their other paper to realise that indeed the incident power density is 20 times higher here than in the other publication. This discussion and explicit comparison needs to be much clearer and included in the main manuscript, ideally on page 9 where some comparisons are made, but much is left to be desired. The authors have to be explicit and upfront about the implications of the gold film 'apparently no longer being needed' after it was apparently crucial for two decades.

Response:

Thank you so much for the useful comments. To better explicit the advantage of ESM, we revised the 1st section in Results, namely the "ESM Setup and Imaging Principles" section, with additional discussions, and new experimental data presented in Fig. 1 and supplementary information.

The key points include:

1, Because of the lower heating effect, ESM allows higher incident intensity than the plasmonic-based single-molecule imaging method, namely the plasmonic scattering microscopy (PSM). The discussion

was presented in the 2nd paragraph in the “ESM Setup and Imaging Principles” section. The temperature-responsive phase transition of the polymer was measured to quantify the heating effect of PSM and ESM (Supplementary Fig. 6). For PSM, we find that the temperature of the gold surface, where the surface plasmons propagate, will reach ~ 62 °C under the incident intensity of 4 kW/cm^2 , which may cause protein misfolding to some extent (Journal of Pharmaceutical and Biomedical Analysis 2020, 189, 113399). So, it is hard to increase the incident intensity further. In contrary, for ESM, the temperature of the glass surface, was less than 33 °C under the incident intensity of 60 kW/cm^2 , which is ~ 20 times higher than the commonly used incident intensity of PSM (Nat Methods 17, 1010–1017 (2020)). This result also indicates that further increasing the incident intensity for ESM will not be limited by the heating effect.

2, ESM allows the short incident wavelength, which can provide a large scattering cross-section. PSM and ESM are constructed on the surface plasmon resonance (SPR) and total internal reflection (TIR) schemes, respectively. The main advantage of SPR supported by a gold film over the TIR for scattering measurement is that SPR can provide ~ 30 times field enhancement, which is ~ 6 times larger than TIR enhancement factor. However, the commonly used gold film SPR scheme requires an incident wavelength of over 600 nm for effective plasmonic excitation, while the TIR allows a wide range of incident wavelengths across the visible spectrum (Supplementary Fig. 7). In this study, we use an incident wavelength of 450 nm for ESM, providing $\sim (660/450)^4$, ~ 5 times larger scattering cross-section than an incident wavelength of 660 nm used for PSM (Fig. 1g, Supplementary Fig. 5, and Supplementary Fig. 8). We also discussed why we do not apply a shorter incident wavelength, because the violet wavelength, such as 405 nm , may destroy the surface modification to interfere with the measurement (Supplementary Fig. 9).

Combining the higher incident light power and shorter wavelength light, we show that ESM can reliably detect single BSA molecule, a medium sized protein (66 kD), while PSM can only reliably detect large protein, such as IgG (150 kD). Therefore, ESM can study a wider range of proteins.

2. I struggle very much with the arguments made in SI Fig N1.3 - which are nevertheless essential to understand the source of contrast. If I read the figure correctly, the authors suggest that there is a difference of a factor of 2 between the scattering from IgA and IgM, which matches a d^3 scaling with diameter. This would suggest that the primary factor for determining contrast is diameter. Yet in Figure 2f they suggest that the signal scales perfectly with molecular weight. For the same two proteins, they report perfect correlation, including a difference in image intensity of a factor of ~ 1.5 for proteins with a mass difference of a factor of almost three. These data contradict each other: what does the signal scale with, mass or diameter? And if it scales with mass, how does the 26 nm PSNP fit in here? I would expect the mass of a 26 nm PSNP to be much, much larger than of IGM, yet the image intensity is only a factor 2 larger? Also, proteins can have very similar mass and very different diameter and vice versa - so what matters and why?

Response:

Thank you so much for the comments, and sorry for the confusion. The revised manuscript uses analyte diameter measured by dynamic light scattering to fit the image intensity. We also discussed the relationship between analyte diameter and image intensity in Supplementary Note 2.

Here are the considerations:

The ESM image of single proteins and small polystyrene nanoparticles arises from the interference of light scattered from an analyte and a reference, which is analogous to the PSM. Rayleigh scattering model can be used to describe the scattering of nano objects. Textbook formulae show that the Rayleigh scattering cross-section σ can be calculated by

$$\sigma = \frac{2\pi^5}{3} \times \frac{d^6}{\left(\frac{\lambda}{n_m}\right)^4} \times \left(\frac{\left(\frac{n_s}{n_m}\right)^2 - 1}{\left(\frac{n_s}{n_m}\right)^2 + 2}\right)^2,$$

where d is the object diameter, λ is the incident wavelength, n_s and n_m are the refractive indices of the object and surrounding media, respectively. The intensity I scattered by the object within an average period of t can be shown as

$$I = |E_s|^2 = \sigma \times P \times t,$$

where P is the incident intensity, and E_s is the amplitude. **These equations show that the diameter, not the mass, directly determines the scattering intensity.**

The equation for scattering cross-section reveals that another factor affecting the analyte image intensity is the analyte refractive index. The proteins on the surface have a refractive index ranging from 1.48 to 1.60, depending on the hydration status (Langmuir 1998, 14, 19, 5636–5648; Meas. Sci. Technol. 17, 932-938 (2005)). On the other hand, polystyrene has a refractive index of 1.61 (refractiveindex.info) under the incident wavelength of 450 nm. Thus, the polystyrene nanoparticle has about 1 ~ 1.8 times higher image intensity than the proteins with the same diameter. The interference image intensity scales with the square root of the scattering cross-section. Considering that the diameter of IgM (21.8 nm) is slightly smaller than the 26 nm polystyrene nanoparticles (Supplementary Table N1.1), the 26 nm polystyrene nanoparticles should have 1.6 ~ 3 times higher image intensity than IgM. Thus, the image intensities of polystyrene nanoparticles can be used to estimate the measurement results of proteins.

In the revised manuscript, we use a new batch of polystyrene nanoparticle with a diameter of 27.9 nm measured by dynamic light scattering, but the discussions maintain similarly (Supplementary Note 2 for details).

As the reviewer stated, the proteins may have similar mass but different diameters, which are caused by different conformations. Some proteins with low molecular weight may present larger image intensity than others with higher molecular weight. The excellent linearity of image intensity against molecular weight should be based on the hypothesis that the proteins have similar conformations. Therefore, we think that diameter should be used to fit the image intensity to avoid confusion. Building the relationship between molecular weight and image intensity requires further study.

3. Fig2 is clearly very important, but also has a number of quite serious issues that must be addressed.

(A) The correlation in Fig. 2f appears perfect. Is this really true? How many measurements were used to obtain the data point and error bar? 1, 2, 3, 5? It appears 1. But in SI Fig N3.2 the authors also show that there is very large variation with cover glass roughness (50% difference in contrast between different measurements). So how did they manage to get such perfect correlation. This very much suggests that the authors conveniently picked histograms. The authors must show several repeats for each protein so that we can understand the true variations in the measurement. The results of each of these repeats should be incorporated in Fig 2f and ideally shown as a box plot including all the data points. In addition, if the relationship is indeed linear as suggested in the figure, I do not understand why the axes chosen to

be logarithmic - the data would be much clearer with linear axes given that we are covering only about 1 order of magnitude in both dimensions.

(B) Fig 2f suggests that they could resolve proteins such as BSA, IgG and IgA given the shown error bars. Yet looking at Figs 2b, c and d, that appears clearly impossible. The error bars do not seem representative. This needs to be checked.

(C) The authors suggest that they are detecting dimers of proteins in all histograms, where they seem to simply add a Gaussian where a dimer would be expected. For this measurement to be meaningful, they need to (a) measure at different concentrations, which should change the amount of dimer; and (b) would need to show several repeats of the same protein to be able to judge to which degree the observed shoulders are statistically significant.

Response:

Thank you for the valuable comments. We have revised the manuscript with substantial new data based on these comments.

(A) About the measurement reproducibility. We picked up the regions with similar roughness profiles on cover glasses to ensure reproducibility in the revised manuscript (Supplementary Fig. 3). Then the repeated measurements show that the image intensity fluctuations are ~5% among different cover glasses. In addition, the measurement results shown in original SI Fig N3.2 are achieved using a polyclonal antibody, which has a large batch-to-batch heterogeneity. In the revised manuscript, we use a monoclonal IgG for measurement. We also use a box plot to show the statistical difference among the image intensities of different proteins (Fig. 2e) in addition to the new calibration curve containing repeated measurement results on three different cover glasses (Fig. 2f).

As discussed in response to the last point and supplementary Note 2, the equations show that the analyte diameter, not the mass, directly determines the image intensity. We use logarithmic scale in Fig. 2f so that we can linearly fit the plot to get the cubical power relationship between the analyte diameters and image intensities from the slope.

As discussed in response to point 7, the signal-to-noise ratio of the current system is close to the limit for measuring protein A. Due to uneven illumination distributions of the current setup, the protein A molecules can only be observed in the central area (Supplementary Figures 12 and 13) with insufficient count for reproducible histograms on some chips. So, we removed the measurement results of protein A molecules from Fig. 2 and showed in the supplementary Fig. 13.

(B) In the revised Fig. 2, we show the result images in grayscale (note that the image intensity scale bars for different proteins are different) and marked the image intensity in the histograms, thus allowing the readers to evaluate the image quality and intensity difference quickly. Besides, we show the dynamic protein binding processes in the same grayscale in Supplementary Video 1, where we can observe the image intensity difference among BSA, IgG, IgA, and IgM. We also use a box plot to show the statistical difference among the image intensities of different proteins (Fig. 2e).

(C) We measured the proteins on different cover glasses (Supplementary Fig. 10), showing that the so-called dimer peak is random. With Increase of the analyte concentration, the dimer peak height does not increase linearly and become a long tail under high analyte concentration (Supplementary Fig. 11). Considering two or more proteins may bind to the surface simultaneously at close locations as discussed in 2nd paragraph in “Detection of single proteins” section, these minor peak and/or tail are highly likely not created by dimers but by two or more molecules simultaneously falling into the same Airy disk area with a diameter of $\sim 1 \mu\text{m}$, thus cannot be resolved by ESM imaging and counted as a single binding event.

4. There is insufficient clarity as to the meaning of the results shown in Fig. 3d-f. The authors suggest that they observe ‘dancing’ molecules, and then go on to characterise the associated kinetics and thereby the affinities. It is not clear how the authors decided where the border between bound and unbound was (it appears about halfway between 0 and the maximum observed intensity) but why? Moving this line up or down would majorly affect the measured kinetics and thus Kd. The authors further state that what they are observing is likely due to surface bound species/a saturated surface. Ultimately, this is an artefact - and I am unconvinced of the value of these measurements for this paper. If they want to demonstrate that they can measure on and off rates in a meaningful way, it appears that they should choose systems with different affinities/measured rates and demonstrate that their measurement agrees, not in the way that it is done here. In addition, are the authors sure that dancing molecules are not simply molecules where the background is fluctuating, and thus the signal is fluctuating?

Response:

We agree with the reviewer that employing halfway between 0 and the maximum observed intensity as standard is not convincing, and we changed the analysis method on the dancing molecules in the revised manuscript.

It should be noted that in revised Fig. 3d-j we employed a published approach to correct the intensity fluctuations resulting from nonuniform background for super resolution lateral localization (Nat. Photonics 13, 480–487 (2019)), which can show the interaction frequency difference more intuitively between specific and nonspecific binding events than the dynamic observation in the video. Thus, the analysis on the different single molecule binding behaviors were moved to Fig. 4.

- (1) Employing the halfway between 0 and maximum observed intensity as the border between bound and unbound conditions has been used for single protein binding kinetics analysis in several publications (Current Opinion in Biotechnology 2011, 22, 1, 75-80; Nano Lett. 2014, 14, 10, 5787–5791). However, it is still unclear about the mechanism of this approach. Thus, we think that it is unsuitable to employ this approach for explaining the experimental phenomenon here. In the revised manuscript, we employed z-axis displacement tracking and free energy analysis based on the statistical distribution of z-axis displacement amplitude to analyze the different single protein binding behaviors. This approach has clear mechanisms and been successfully used for analyzing the adhesion bond conditions (details shown in Supplementary Note 4; Ref: JACS 2019, 141(40), 16071–16078; PNAS 2003, 100 (20), 11378–11381).
- (2) In the revised manuscript, we analyze the statistical distribution of z-axis thermal fluctuations of one IgM molecule under two absorption conditions on the antibody modified surface, namely ‘binding’

and 'dancing', as shown in Supplementary Video 3. Both the Supplementary Video 3 and z-axis displacement tracking result shown in Fig. 4b show that the IgM molecule under 'dancing' condition has much larger thermal fluctuations than 'binding' condition. Then, the statistical analysis shows that the anti-IgM/IgM adhesion bond has ~10 times smaller effective spring constant under 'dancing' condition than 'binding condition', reflecting the restoring force decreases by ~10 times during changing 'binding' to 'dancing'. In addition, the 'binding' and 'dancing' phenomenon was observed on the same IgM molecule, so we think that this is not an artefact. We also track the thermal fluctuations in the adjacent area, confirming that the thermal fluctuations observed in Fig. 4a-d were not caused by background fluctuations (Supplementary Fig. N4.1).

- (3) Commercial antibody usually has a very high affinity, and the behaviors of different proteins on the corresponding antibodies are hard to be recognized. To show the meaning of single analyte binding behavior analysis, we employed single-stranded DNA (ssDNA), which will change from a soft chain to rigid structure after hybridization with complementary microRNA, as an example in the revised manuscript (Fig. 4 e-i). Gold nanoparticles were linked onto the ssDNA, and the nanoparticle z-axis thermal fluctuations decreases after hybridization with complementary microRNA (Fig. 4e), demonstrating that ESM can determine the properties of adhesion bonds created by molecular linkages. The statistical analysis further supports the DNA conformation changes can be monitored by ESM. This also provides a label-free detection scheme for small molecules, such as miRNA, which is challenging for the evanescent based measurement approaches due to their small sizes (Nature Photon 11, 477–481 (2017); Sensors and Actuators B: Chemical 347, 130629 (2021)).

5. I am surprised to see the results in Fig. 4: The results shown in Fig. 4b suggest that the authors can achieve sub-nm localisation precision, which is what would be required for the dots to be meaningful, and thus the extraction of diffusion coefficients to be meaningful. I do not understand how this could possibly be the case given the highly non-uniform PSF of the method. In addition, I am very surprised by the very low diffusion coefficient observed - it would seem that the motion of such a small molecule on such a small tether would completely average out on the timescale of the measurement. As previously, how reproducible are these results? It appears that the data shown is from a single assay. How do the distributions in c,d and e vary from assay to assay, with different surfaces, on different days? One could completely explain Fig. 2b by a slight change in localisation precision/fitting quality/focus drift after addition of mRNA. Have the authors done controls where they flowed in buffer not containing miRNA to see if that affects the measurement? Are the differences before and after addition of miRNA reproducible for multiple experiments? In addition, I could not find any information on how the tracking was actually done - which seems essential. I would be surprised if feeding the observed PSFs into track mate could provide any meaningful results whatsoever.

Response:

We agree that the Fig. 4 in original manuscript presents the results from a non-rigorous experiment. We have revised the experimental designs and presented new data in the revised manuscript.

In the revised Fig. 3, the lateral localization errors differ from protein to protein even after correcting the intensity fluctuations caused by nonuniform background. As a result, the lateral tracking results shown in old Fig. 4 should be unconvincing. **Thus, in the revised manuscript, we employ z-axis displacement tracking analysis, which could provide high tracking precision due to the exponential**

decaying properties of evanescent field intensity, for evaluating the properties of linking DNA. The tracking workflow was described in Supplementary Note 4, and can also be found in the literature (JACS 2019, 141(40), 16071–16078).

As discussed in previous point, in the revised Fig. 4a-d, we analyze the thermal fluctuations of one IgM molecule on anti-IgM modified surface under different molecular interaction modes. The comparison of effective spring constant, which reflects the restoring force of molecular adhesion bond and is subtracted from the statistical distribution of z-axis thermal fluctuations, shows that the adhesion bond resulting from molecular binding relaxes when IgM changes the ‘binding’ condition to ‘dancing’ condition. This demonstrates the z-axis displacement tracking with ESM can be used to evaluate the adhesion bond thermal fluctuation properties.

Then, we link the 40 nm gold nanoparticles to the ssDNA immobilized on the surface. The nanoparticle intensity was strong, and independent of background (Supplementary Fig. 15). Then, the nanoparticle z-axis fluctuations were tracked and analyzed for comparison, and the effective spring constant of DNA linker was measured and found to be increased after hybridization of ssDNA with complementary microRNA (Fig. 4e-g). Considering that the effective spring constant reflects the restoring force of adhesion bond, this indicates that the DNA linkers become rigid, agreeing with the fact that the soft ssDNA will become a more rigid structure after hybridizing with complementary microRNA.

To show the measurement reproducibility, the statistical analysis on 64 of gold nanoparticles on another cover glass also shows that the effective spring constant response to the DNA conformation changes (Fig. 4 h-i). More repeated measurements on different cover glasses were shown in Supplementary Fig. 16. In addition, the force analysis on the nanoparticle linked by DNA also shows that the nanoparticle motion should be dominated by thermal fluctuations of adhesion bond, rather than the diffusion (Supplementary Note 5). The background analysis demonstrates that the experimental conditions were stable during the measurement (Supplementary Note 4).

The measurement was performed under a constant flow rate to maintain a stable measurement conditions, and the solution change was performed with an injection valve without varying the flowing rate. The ssDNA and rigid duplex structure linked nanoparticles were measured in the same PBS buffer to exclude the possible heterogeneity caused by different buffers.

6. I do not understand, and the authors would need to clarify, how it is possible that supposedly the signal intensity is independent of local image background (SI Fig 4), but dependent on overall image intensity (SI Fig N3.2)? The author clearly state and suggest that their measurement is interferometric, combining scattered and background light. How then, can they generate the same signal when the protein lands in a dark vs a bright region of the glass? That seems completely impossible.

Response:

We removed this claim in the revised supplementary information. This is a complicated phenomenon. Although some theoretical explanation on this issue already exists as presented in the main text, a robust model for this phenomenon requires further study.

The phenomenon of analyte signal intensity being independent of background intensity for evanescent scattering imaging was firstly found on the plasmonic scattering microscopy (Nature Methods 17, 1010-1017 (2020); ACS Sensors 6, 1357-1366 (2021)). The evanescent waves scattered by the objects can be

divided into two kinds (Phys. Rev. B 92, 245419, 2015). One is elastic scattering, which can be observed by monitoring the reflection beams (Angewandte Chemie International Edition 58, 572-576 (2019)). The other one is out-of-plane scattering, which ESM and PSM observe. The fact that they are related to each other makes the phenomenon complicated (Optics Express 17, 15, 12470-12480 (2009)). The elastically scattered evanescent waves are delocalized and can interfere with each other to redistribute the illumination field (Physical Review B 58, 10899-10910 (1998); Anal. Chem. 2014, 86, 18, 8992–8997). Thus, the interference of evanescent waves scattered by the high-density surface roughness may create a comparable homogenous reference field. But a robust model requires quantifying both the intensity and phase.

The Foreman group recently presented the theoretical analysis on the phase statistics of the evanescent scattering imaging utilizing the interference of surface roughness with the analyte. They theoretically proved that the phase difference is small because of the short distance between scattering sites of surface roughness and analyte binding positions (ACS Photonics 8, 2227-2233 (2021)). In addition, the phase statistics are identical within the field of view because the phase difference is random in the multiple scattering regime (Physical Review Research 3, 033111 (2021)). These discussions and literature were added in the context following Equation (1) in the revised manuscript. They can explain the effect of phase distribution on the image intensity measurement, but the impact of illumination evanescent field redistribution still requires more studies to be quantified. Therefore, we remove this claim to avoid confusing the readers.

7. I struggle to understand some of the observations in the supplementary videos. It appears that the authors use a relative large illumination area (although I could not see it defined). Why is it that Protein A only binds in a 2x2 micron area, BSA in a 4x4 micron area, and IgA apparently everywhere? This is very confusing indeed and needs to be clarified/discussed.

Response:

Thanks for the insightful question.

For the definition of field of view: We present the raw image to show that the field of view is defined by illumination area (Fig. 1b).

For BSA binding: For results reported in the revised manuscript, we prepared the surface modification on cover glasses more carefully, specifically we removed the water residue as much as possible before the amino group modifications in each batch. Thus, in the revised Supplementary Video 1, we can find that the BSA binding events can be observed everywhere within the field of view as other proteins, like IgA. Therefore, we are convinced that the BSA binding issue in the original manuscript is associated with cover glass modification efficiency in some batches ((Langmuir 22, 11142-11147 (2006)).

For the protein A analysis: We add the image intensity analysis at different areas within the field of view and find that the center area has slightly higher image intensity than the surrounding area (Supplementary Fig. 12), which likely caused by the Gaussian distribution of laser beam. Thus, the protein A, which is among the smallest molecules detected using label-free single molecule systems, can only be observed in the central area due to the stronger incident intensity for higher signal to noise ratio, which is a 2x2 micron area. Considering that the protein A cannot be imaged within the whole field of

view, we have removed the measurement results of protein A from Fig. 2 and presented them in Supplementary Fig. 13.

8. The colour scale chosen for the images in the main manuscript (blue to orange) is very poor. It makes it near impossible to see/judge the image quality. This should be optimised.

Response:

We used gray scale differential images in the revised manuscript for easily judging the image quality.

Reviewer #2:

Remarks to the Author:

P. Zhang et. al. developed the evanescent scattering microscopy (ESM) based on total internal reflection (TIR) concepts to perform single-molecule imaging on plain glass surfaces. The authors have demonstrated their technique by quantifying the mass of single proteins, studying the ligand interactions, and measuring the diffusion properties of single molecules. I do not recommend publishing this manuscript mainly because of the following two reasons,

Response:

Thank you for your comments. We added new experimental results and discussions to show the advantages of ESM more clearly in the revised manuscript as detailed below each of questions.

1) These concepts and experiments carried out in this manuscript are similar to their previous paper [Nature Methods 17, 1010–1017 (2020), Reference 21 in the manuscript] where the authors developed plasmonic scattering microscopy by utilizing surface plasmon resonance (SPR) concept. The key difference in this manuscript is that instead of using SPR, TIR was used with the glass surface. The optical setup, image processing, and even the experiments are very similar to the previous paper. I agree with the authors that using glass surfaces will inhibit plasmonic quenching and heating, however, I don't believe this as a novel advancement to report in Nature Communications. In addition, no new scientific insights have been obtained on the single-molecule studies as compared to their previous paper as well as to the other single-molecule imaging reports.

Response:

Thanks for the comments. We have added substantial new experiments and discussions to demonstrate the advantages of the evanescent scattering microscopy (ESM) over PSM.

First, we present the experiments to quantitatively show that the heating effect limits the applications of plasmonic scattering microscopy (PSM). High incident light intensity is usually required to achieve sufficient signal to noise ratio for single molecule optical imaging techniques. For the PSM approach reported in the Nature Methods, we find that the surface temperature reaches ~62 °C under the incident intensity of 4 kW/cm² by observing the response of temperature sensitive phase transition materials on the surface (Supplementary Fig. 6). This temperature could vary the characteristics of some

proteins (ACS Sens. 2021, 6, 3, 1357–1366), thus limiting the widespread applications of PSM. This also agrees with the report that the large plasmonic enhancement usually accompanies with strong heating effect (Nature Nanotech 10, 25–34 (2015)). So, we think that it is necessary to develop an evanescent single molecule imaging approach without plasmonic heating effect for broader applications. This consideration is discussed in 2nd paragraph in “ESM Setup and Imaging Principles” section of the revised manuscript. We show that without the plasmonic heating effect, ESM have 20 times higher incident light power to further reduce shot noise.

Second, ESM can employ different incident wavelengths for different experiments (Supplementary Fig. 8), such as employing short incident wavelength for large scattering cross section of dielectric analytes (Fig. 1-3), and employing green incident wavelength for exciting localized SPR of gold nanoparticles for large scattering signal (Fig. 4 and Supplementary Fig. 14). In contrast, PSM usually only permits incident wavelength longer than 600 nm (Supplementary Fig. 7). With a high-power short incident light (60 kW/m² at 450nm), we show that ESM can reliably detect single BSA molecule, a medium sized protein (66 kD), while PSM can only reliably detect large protein, such as IgG (150 kD). Therefore, ESM can study a wider range of proteins than PSM. Furthermore, without the gold layer absorption, ESM also provides the flexibility for coupling with other imaging techniques, such as fluorescence and Raman imaging, leading to multi-modal imaging information.

Third, in the protein binding kinetic analysis, we correct the intensity fluctuations caused by nonuniform background, and calculate the super resolution localizations of protein binding sites to provide the interaction frequency difference information between specific and nonspecific binding events, which are more intuitive than only dynamically observed in the video shown in PSM (Fig. 3).

Finally, we presented the applications of ESM for detecting small molecules, and use microRNA as an example. We employ the ssDNA linked gold nanoparticles as detection system, and tracking the nanoparticle z-axis thermal fluctuations for detecting the microRNA binding. This system also was reported by SPR, but the electron transfer between gold nanoparticles and the gold film will lead to the nonlinear variations of SPR scattering signals, making it challenging to be realized with conventional optical system and image processing approaches (Fig. 4 and 3rd paragraph in Discussion section).

This work is not a simple extension of plasmonic scattering microscopy, but it demonstrates a solution for developing evanescent single molecule imaging approach without plasmonic enhancement and its associated side effects. We think that ESM will become an important tool complementary to PSM for single molecule imaging.

2) The references were not properly cited, it looks like authors have given more importance to the journal name/IF/publishing group more than the actual science itself. First, since the manuscript is submitted to Nature Communications, 46% of the cited references are from Nature journals. Second, many of these references look like they were forcibly cited, I would like to give two examples – a) Ref 1 has nothing to do with single-molecule detection or biological interaction, and b) In page 4, the authors mentioned that they have used TrackMate plugin for analysis and cited Ref 34 and 35, in which 35 is relevant whereas 34 does not discuss the specific plugin. If authors want to cite the ImageJ, they can cite the website directly.

Response:

Thanks for the careful reading! In the revised manuscript, we cited the literatures more specifically. For the ref. 1, it was wrong due to the reference management software bug. We also removed the Nature Methods paper about Fiji, and only keep the TrackMate reference.

Reviewer #3:

Remarks to the Author:

I find the paper interesting and worthy of publication. The technique proposed and demonstrated offers important advantages and should be of interest to the single molecule sensing community. However, the writing of the paper needs improvements.

Response:

Thank you very much for the positive feedback on our work. We have addressed the comments by new experiments and discussion as detailed below each of the questions.

Major points: Please respond and make amendments to the paper accordingly. (Text reproduced from the manuscript is put in blue color.)

1. The main innovation in the experimental technique is using the interference of light scattered by natural surface roughness of the glass slide and light scattered by molecules immersed in the evanescent fields near a glass interface when illuminated from the glass side at an angle of incidence larger than the critical angle. However, little is said about the roughness parameters of the interface, other than presenting an atomic force line scan (supplementary Fig. 2) and that the SNR is independent of the roughness parameters. A discussion about the requirements of the roughness is needed in the main text. Is it just a happy coincidence that bare glass-slides have the appropriate roughness? Can it be intentionally improved? On what does the phase difference, θ , depends? Is the same for all pixels in the images? How does it depend on the position of the molecule, on the roughness and the viewing angle?

Response:

Thanks for the comments. We have revised the manuscript accordingly to address these issues.

About the requirements of the roughness. The cover glasses have variable roughness features in the commercial batch. We picked up the regions on cover glasses with suitable roughness to ensure measurement reproducibility (Supplementary Fig. 3). In future, we think that the surface roughness can be intentionally improved by coating additional films. According to our previous paper on plasmonic scattering microscopy (ACS Sens. 2021, 6, 3, 1357–1366), the gold-coated glass slide prepared by thermal evaporation can suppress the measurement fluctuations to be smaller than 3%. Other transparent film coating except Au can be considered in the future to reduce the roughness difference on cover glasses from batch to batch.

About the phase difference distribution. Recent research reports the phase difference distribution under multiple scattering systems, such as plasmonic scattering microscopy and evanescent scattering microscopy reported here. The phase difference is small because of the short distance between scattering sites of surface roughness and analyte binding positions (ACS Photonics 8, 2227-2233 (2021)), and the phase statistics are identical within the field of view because the phase difference is random in the multiple scattering regime (Physical Review Research 3, 033111 (2021)). The discussions were added in the context following Equation (1) in the revised manuscript.

2, Figure 1 in the main text is essential to understand the paper and needs more detail. May be supplementary Fig. 1 should be part (b) of figure 1 in the main text. In the current Fig. 1 it appears that a

well collimated laser beam is incident at a well-defined angle of incidence. However, in supplementary Fig. 1 one can see that light is focused on the interface, and light is incident at a range of angles of incidence around 65 degrees. This means that there is evanescent light with different penetration lengths illuminating the other side of the interface.

Response:

We revised Fig. 1, Supplementary Fig. 1, and their captions.

In the experiment, the laser beam was focused on the back focal plane of the illumination objective. Then the collimated laser beam was directed onto a cover glass mounted on the objective via refractive index match oil. So, the laser beam was collimated and illuminating the field of view with the configured incident angle. The revised figures show this system configuration more clearly.

In addition, the revised Fig. 1 also presented the raw and differential images when measuring analytes, image intensity versus analyte size, and image intensity versus incident wavelength to help readers understand the imaging mechanism more intuitively.

3, Related to the latter point, it appears to me that the angles of incidence (since it is not a single angle of incidence due to focusing) is a crucial parameter that needs to be set with care. The penetration length depth depends on the angle of incidence. Some discussion about this is given in the supplementary notes. However, something must be said in the main text.

Response:

We revised Fig. 1 to show the experimental configuration, where the laser beam was collimated and illuminating the field of view with the configured incident angle, more clearly.

We add the description of the incident angle at the 4th line in the “ESM Setup and Imaging Principles” section in the revised manuscript. The incident angle was fixed in all experiments to ensure stable penetration depth of the evanescent field. We also add figures and more detailed descriptions of evanescent penetration depth calculations in Supplementary Note 1.

4, It is not clear to me what does the intensity histograms in Fig. 2 mean and how they must be interpreted. What determines the “total intensity” in an image at a protein binding site? Why some molecules produce one total intensity and other molecules produce other total intensity?

Response:

Intensity histograms reflect the size distribution and scattering cross-section difference of various nanoparticles and proteins. According to scattering cross-section equation shown in Supplementary Note 2, scattering intensities of different nanoparticles are different, result in different image intensities. In addition, the nanoparticles and proteins have diameter variations due to their intrinsic properties. The dynamic light scattering shows that the nanoparticles have diameter variations of 10~20% (Supplementary Fig. 5), and proteins have diameter variations of ~2 nm around the mean diameter.

To help readers understand how the ‘total intensity’ is generated, we add the image processing steps in Fig. 1c & 1d. To achieve the image of small analytes, such as a 27.9 nm polystyrene nanoparticle, the raw

images recorded by the camera should first be subtracted by the previous frame to achieve a differential frame. Then, we can see that one bright spot appears on the differential image due to the analyte binding on the surface. The scattered light from the analyte will interfere with the background, leading to varied image intensity at binding locations (Equation 1). The image intensity variation is the “total intensity” in the previous version. Limited by the optical diffraction, the bright spot has the Airy disk diameter, estimated to be $\sim 1.07 \mu\text{m}$ after dividing the incident wavelength by the imaging objective numerical aperture. Its intensity scales with the analyte size (Fig. 1e). We determine the image intensity by averaging the powers of all pixels within the Airy disk.

Minor, but mandatory, points: The following sentences are not clear and need revision. (Text reproduced from the manuscript is put in blue color)

1. The words “diffraction limit” are used through the paper. A clear definition of what it is meant by this is needed. What is its value in the current experimental setup?

Response:

The revised manuscript replaced the diffraction limit with Airy spot, whose diameter is limited by diffraction. We add the definition of Airy spot for this study in the 14th line in the text following Equation (1). The diameter of the Airy disk was estimated to be $\sim 1.07 \mu\text{m}$ after dividing the incident wavelength of 450 nm by the imaging objective numerical aperture of 0.42.

2. Top of page 2: “Single molecule detection is required to analyze the heterogeneous and stochastic processes for deep understanding of mesoscopic scale biological interactions at the level of detail.” Specifically, what it is meant by “heterogeneous process” and by “scale biological interactions at the level of detail” Not clear.

Response:

To avoid confusion, we rephrased this sentence as “Single-molecule detections push beyond ensemble averages and reveal the statistical distribution of molecular sizes and binding processes.”

3. Near the bottom of page 2: “..wide field imaging applications” What are wide field applications?

Response:

To avoid confusion, we rephrased this sentence as “it is still challenging to employ these exquisite microspheres and nanomaterials for wide-field single-molecule imaging applications, such as parallelly monitoring the dynamic molecular binding process in different locations.”

4. Middle of page 3: “In addition, we show that ESM can provide high resolution images by eliminating the parabolic tails, which is a common issue for the traditional evanescent imaging system” What are “parabolic tails” please clarify.

Response:

We removed this sentence in the revision. Parabolic tails are the unique V shaped nanoparticle scattering pattern observed in SPR and TIR imaging.

5. Figure caption of supplementary Fig. 3 is not clear to me: “The raw image sequence of a bare cover glass was recorded at 200 frames/s for 30 s. Then, differential image sequences with different image average times were obtained. Next, total intensity of all pixels in a selected airy disk sized spot (~6 pixels diameter circled area) for all images in the differential image sequences were calculated. Standard deviation of the total intensity in each 30 s differential image sequences are calculated and plotted as black dots. The red line is expected shot noise value based on the total photons collected in the spot. Therefore, for the averaging time below 50 ms, shot noise is dominant. Incident light intensity, 60 kW cm⁻². Exposure time, 5 ms.” What is it meant by: “raw image sequence”, “Differential image sequence”, “total intensity” and “Airy disk size spot” Looking at the main text it says “Then, differential images were achieved by subtracting a previous frame from each frame to remove $|E_b|^2$.” Shouldn’t we get zero?

Response:

We revised the original Supplementary Fig. 3 and renamed it as Supplementary Fig. 4 in revised version. The raw image sequence contains the images directly recorded by the camera (Supplementary Fig. 4a).

As discussed in response to Major point 4, each raw image frame should be subtracted by the corresponding previous frame to achieve a differential frame, in order to observe the binding of small objects. The differential image sequence was acquired from the raw image sequence with differential processing. However, the background in the differential image is not zero due to the shot noise, which is intrinsic and scales with the square root of the photon number (Supplementary Fig. 4b). The shot noise can be suppressed by image averaging, while the mechanical drift limited the longest average period (Supplementary Fig. 4b and 4c).

The bright spot created by objects on the differential images has the Airy disk diameter, which is limited by optical diffraction and estimated to be $\sim 1.07 \mu\text{m}$ after dividing the incident wavelength by the imaging objective numerical aperture. The image intensity of the bright spot is the “total intensity” in the previous version.

6. In caption of supplementary Fig. 4 the sentence: “The small second peak (blue line) is attributed to formation of dimers or two particles binding to the nearby surface simultaneously with distance smaller than the diffraction limit.” is confusing. Clarify “distance smaller than the diffraction limit”. Also, the word “corrcoef” near the end of the caption must be corrected.

Response:

The original supplementary Fig. 4 was removed in the revised version to avoid confusion after considering reviewer 1’s point 6.

For “distance smaller than the diffraction limit.” As discussed in the latter point, the nanoparticle binding creates the bright spot with an Airy disk diameter of $\sim 1.07 \mu\text{m}$, which is limited by the optical diffraction. This diameter was much larger than the nano-objects. Therefore, if two nano-objects bind to the surface with a distance smaller than $1.07 \mu\text{m}$ simultaneously (on the same image frame), we can only observe a combined bright spot as a single binding event, not two independent bright spots.

The “corrcoef” is a MATLAB function for correlation coefficient calculation.

7. In the appendix of supplementary note 1: Provide a reference for the equation. Explain how this equation is obtained. Define E_{sp} . How are effective diameters calculated? Some of these ideas must be mentioned in the main text of the paper.

Response:

We mentioned this consideration in the 9th last line at the 1st paragraph in the “ESM Setup and Imaging Principles” section.

We rephrased Supplementary Note 1 to discuss the effective diameter calculation. We add the reference (PNAS, 2010, 107(37): 16028-16032) and the model figure (Supplementary Fig. N1.1) to show the considerations for calculating the effective diameter intuitively. The “ E_{sp} ” in the previous submission was used to represent the effective diameter. This isn't very clear and has been replaced by D_{eff} in the revised supplementary information.

8. In supplementary Note 4: How is the equation provided obtained? What is it meant by “Tracking precision”? σ is the standard deviation of what? How is it possible to obtain particles' displacements in the order of 1 nm with an image of about 6 microns wide?

Response:

The original Supplementary Note 3, namely chip to chip reproducibility, has been replaced by Supplementary Fig. 5 and 10 to show the additional experiments, so the old Supplementary Fig. 4 has been updating to Supplementary Note 3 in the revised version. We also revised Fig. 3 to show the super-resolution localization workflow.

The equation is from the super-resolution localization of fluorescent dye job (Equation 1 in Science, 300, 2061-2065 (2003)). For a Gaussian distributed bright spot, such as an Airy spot created by optical diffraction in optical imaging, the position for its centroid can be confirmed by Gaussian fitting. Due to the intensity fluctuations created by shot noise, the position recognized by fitting differs against time. The standard deviation of the position value sequence achieved at different times indicates the precision determining the lateral position of one object with this approach (Fig. 3g). This is also the basis of super-resolution fluorescence imaging, where the fluorescence dyes flash randomly, and their positions were determined by Gaussian fitting to achieve ~10 nm resolution. The ~1 nm resolution was overclaimed in the old version. We have rechecked the resolution by monitoring more molecules for statistical analysis, showing that IgA molecules' localization precision is ~16 nm (Fig. 3h). This can be used to achieve the super-resolution localization of molecular binding sites to evaluate the binding frequency intuitively (Fig. 3i and 3j).

9. Supplementary Note 6: Please cite references for the equations used.

Response:

The original supplementary Note 6 has been updated as supplementary Note 5 in revised supplementary information. We have listed the references for each equation. Among them, for the textbook equations, we cited the wiki links, and for others, we cited the original and following publications.

Reviewers' Comments:

Reviewer #1:

Remarks to the Author:

The authors have made substantial changes to the manuscript in the context of extensive feedback by the referees. Judging by the colouring of the manuscript, more than 90% of the results section has been rewritten. It appears that both Figs 3 and Fig 4 are essentially completely new. My main concern is that the authors introduce a method, which has been substantially altered in terms of its capabilities and analysis approach, in Fig.2 and then use it in Figs 3 & 4, but without the method being properly verified. While I appreciate the authors' drive towards applications of their methodology, the current evidence provided for how the method works, what it depends on, how it is implemented and how it compares to other methods is insufficient to support the applications presented. I would urge the authors to go through a much more thorough process and thus be convincing about the basics of single molecule detection and quantification before moving on to applications. I will illustrate these issues below:

1. The authors have dropped their claim of detecting Protein A, owing due to low SNR. What is very surprising in the new manuscript is that all protein images in Fig2 and in the SI now appear to be perfect 'bright spots' almost like in fluorescence microscopy, as opposed to the interferometric PSFs we had in the past. How is that possible?
2. I do not understand the relationship between the images shown in Fig2 and the histograms. The signals for BSA in the shown images is about 20 counts at the peak, but the histogram has a mean of 2.2. It is not clear how these are related. How does one observe a signal of '2' given the noise shown in the BSA image?
3. Why do the authors now show SI Fig N2.1 in the main manuscript? It appears to be a critical connection between Figure 1 and 2, and would need quantitative analysis to convince readers that indeed they are seeing single molecules.
4. The authors have dropped their argument on seeing a mass-relationship, and now argue that the only thing that matters is 'diameter' (response to point 2). This is further confirmed by SI Fig N2.1, where they show a perfect diameter scaling for proteins and nanoparticles. However, they seem to ignore the importance of refractive index of the object, which is immensely different for a nanoparticle than for a protein, especially at the interface between 1.52 (glass) and water (1.33). One cannot reduce the scattering cross section equation to being proportional to diameter while there is a ns/nm term, which is close to zero for protein, and far from zero for polystyrene according to the refractive index numbers given by the authors.
5. The authors claim that their signal is interferometric, so it should scale with the scattering amplitude rather than cross section. In this sense, it should scale identically to the scaling shown for iSCAT or mass photometry, which has been shown to be proportional to mass. Why is it different here, if we have essentially the same process going on (just a different detection methodology). If the signal is interferometric, why is always positive? Can it be negative?
6. I very much struggle with the fundamental feasibility of their experiment. Using their equation for the scattering cross section, I obtain a cross section of about 10^{-11} um^2 for BSA (compare also with Piliarik and Sandoghdar Nature Communications 2014). This would amount to about 60 photons scattered per 5 ms in total, of which no more than 10% (and probably less with a 0.4 NA) would be detected in their approach (because most photons travel towards the higher index material. This would be impossible to detect even with the world's best camera (once distributed over a realistic PSF). According to these estimates, the authors would require about 50 times more power to record the kind of image of BSA that they show in Fig. 2. The images they show in Fig. 2 appear physically impossible to record.

Reviewer #3:

Remarks to the Author:

The paper has improved noticeably clarifying all the main issues. It is easier to read and more detailed explanations are given. All my queries were answered satisfactorily. Overall, the paper is reporting very interesting high-quality work. Yet still the writing should be improved. The paper still has many grammatical errors, and several sentences are not well constructed, making it harder to understand. Nevertheless, with some patience it can be understood.

After some English editing throughout, I recommend publication.

A note to the authors' response letter: Besides stating their response to each of the comments, in all cases, the authors should explain in detail the corresponding changes done the paper and pinpoint their location of such changes in the revised text; preferably reproducing the amended text just after their response to the reviewer. If no changes were done after a given comment, they should say so and justify it.

Point to Point Response to Reviewer's Comments

We are grateful for the positive feedback from the reviewers! Following the advice, we have added quantitative analysis, experimental results, and related references to address all the questions from reviewer 1. The revisions on the manuscript and supplementary information are marked in blue color. We provide point-to-point responses after individual questions that quoted in italic.

Reviewer #1:

The authors have made substantial changes to the manuscript in the context of extensive feedback by the referees. Judging by the colouring of the manuscript, more than 90% of the results section has been rewritten. It appears that both Figs 3 and Fig 4 are essentially completely new. My main concern is that the authors introduce a method, which has been substantially altered in terms of its capabilities and analysis approach, in Fig.2 and then use it in Figs 3 & 4, but without the method being properly verified. While I appreciate the authors' drive towards applications of their methodology, the current evidence provided for how the method works, what it depends on, how it is implemented and how it compares to other methods is insufficient to support the applications presented. I would urge the authors to go through a much more thorough process and thus be convincing about the basics of single molecule detection and quantification before moving on to applications. I will illustrate these issues below:

1. The authors have dropped their claim of detecting Protein A, owing due to low SNR. What is very surprising in the new manuscript is that all protein images in Fig2 and in the SI now appear to be perfect 'bright spots' almost like in fluorescence microscopy, as opposed to the interferometric PSFs we had in the past. How is that possible?

Response:

We add the discussion at the 1st line after Equation (1). "The phase difference determines whether the interferometric contrast, namely the $2|E_b||E_s|\cos(\theta)$, is negative or positive. Unlike the interferometric scattering, where the Gouy phase shift dominates and leads to a negative interferometric contrast^{11,22}, the phase difference is close to zero in ESM because of the short distance between scattering sites of surface roughness and analyte binding positions, leading to a positive interferometric contrast as the plasmonic scattering microscopy^{17, 18, 22, 23}".

The key point is that the phase determines whether the interferometric contrast is positive or negative, namely whether the spots are bright or dark. For the interferometric scattering (iSCAT) system, the Gouy phase shift brings the phase shift of π , making the image contrast negative. For the previously reported plasmonic scattering microscopy and evanescent scattering microscopy reported here, the proteins and surface roughness are very close to each other. Thus, the phases of their scattering are identical, resulting in positive image contrast. A group from Imperial College has theoretically proved this (ACS Photonics 8, 2227-2233 (2021); Physical Review Research 3, 033111 (2021)), and this has also been experimentally demonstrated by our evanescent plasmonic scattering microscopy (Nature Methods 17, 1010-1017 (2020); ACS Sensors 6, 1357-1366 (2021); ACS Sensors 6, 4244-4254 (2021)).

In addition, we should point out that the PSF of ESM reported by us is not perfect. We add representative line profiles of a polystyrene nanoparticle and a BSA protein in Supplementary Fig. N3.1. The bright spot has asymmetrical intensity distribution, and some pixels included by the bright spot have negative intensities, similar to the results achieved by evanescent plasmonic scattering microscopy

(Nature Methods 17, 1010-1017 (2020); ACS Sensors 6, 1357-1366 (2021)). This pattern is more complex than the Airy pattern, so we use mean intensity, rather than maximum intensity, as the sensor output.

2. I do not understand the relationship between the images shown in Fig2 and the histograms. The signals for BSA in the shown images is about 20 counts at the peak, but the histogram has a mean of 2.2. It is not clear how these are related. How does one observe a signal of '2' given the noise shown in the BSA image?

Response:

We added the explanation about the intensity scale and signal histograms at the 15th line in the first paragraph in the 'Detection of single proteins' section. "The maximum value of image contrast scale was set to be 1.5 ~ 2 times higher than the maximum intensity of the bright spots created by the proteins on the image for easy reading, and the mean value of the intensities of all pixels included by the bright spots was used to construct the histograms for evaluating the signal intensity more precisely (Supplementary Note 3).".

We also added the detailed explanation in Supplementary Note 3. The negative pixel intensity within the bright spots, blank pixels taken by the TrackMate, and camera pixel noise make the maximum intensity 5~7 times larger than the calculated mean intensity while analyzing the bright spots achieved by ESM. Considering that the maximum value of the image contrast scale was set to be 1.5 ~ 2 times higher than the maximum intensity of the bright spots to achieve a comfortable contrast for easy reading, we can see that the maximum value of the intensity scale is ~10 times larger than the mean intensity shown in the histogram in Fig. 2.

3. Why do the authors now show SI Fig N2.1 in the main manuscript? It appears to be a critical connection between Figure 1 and 2, and would need quantitative analysis to convince readers that indeed they are seeing single molecules.

Response:

We have added quantitative analysis in Supplementary Note 2 to convince the readers that we see single proteins. The quantitative analysis shows that the ESM measurement signal-to-noise ratio is ~10 for single BSA proteins, agreeing with the experimental results. In addition, we should point out that the evanescent field created by total internal reflection has 5x intensity enhancement, and averaged period of 50 ms is employed for image processing. In point 6, the reviewer pointed out that a 50 times higher incident intensity is needed to achieve the signal-to-noise ratio we reported based on the exposure time of 5 ms without considering the evanescent field enhancement. Thus, combining the 50 ms (10 frame) average period and 5 times evanescent field enhancement, the 50 times higher power intensity requirement is met. The feasibility of using evanescent field enhancement for improving signal-to-noise ratio has been demonstrated by the PSM (Nature Methods 17, 1010-1017 (2020)) and one iSCAT group from the US (Nat Commun 12, 1744 (2021); Nat Methods 18, 447 (2021)), and the approach of using image average to improve the signal-to-noise ratio is also widely used by iSCAT groups (Nat Commun 5,

4495 (2014); Science 360, 423-427 (2018)). We emphasize these two factors in the 2nd line in the 2nd paragraph in the 'ESM Setup and Imaging Principles' section.

In the main text, Fig. 1 is used to show the reason for setting experimental conditions by employing the polystyrene nanoparticles as samples, such as the construction of optical setup and incident wavelength, and Fig.2 is used to show the results of imaging single proteins. We want to ensure that the readers focus on measuring single proteins with the current version of Fig. 2.

4. The authors have dropped their argument on seeing a mass-relationship, and now argue that the only thing that matters is 'diameter' (response to point 2). This is further confirmed by SI Fig N2.1, where they show a perfect diameter scaling for proteins and nanoparticles. However, they seem to ignore the importance of refractive index of the object, which is immensely different for a nanoparticle than for a protein, especially at the interface between 1.52 (glass) and water (1.33). One cannot reduce the scattering cross section equation to being proportional to diameter while there is a ns/nm term, which is close to zero for protein, and far from zero for polystyrene according to the refractive index numbers given by the authors.

Response:

For interferometric imaging of proteins and small nanoparticles, the ESM image intensity versus diameter obeys the cubic law, not linearly proportional to diameter. To clarify, we added notes of logarithmic scale plot to Fig. 2f and Supplementary Fig. N2.1 in the captions in the revised manuscript and supporting information.

We also add the discussion about the refractive index difference between polystyrene and proteins in the 2nd paragraph in Supplementary Note 2. The quantitative analysis shows that the refractive index difference between polystyrene and proteins will result in up to a 20% size determination error. However, it has been pointed out that the surface charges of proteins will enhance their scattering in the evanescent field (Nature Photon 11, 477-481 (2017); Nat Commun 11, 4768 (2020)). Thus, the polystyrene nanoparticle can be directly used to estimate the protein diameter for evanescent imaging (Nat Commun 11, 4768 (2020); Nat Methods 17, 1010-1017 (2020)). Supplementary Fig. N2.1 also shows that the image intensity of 27.9 nm polystyrene nanoparticle is also close to the expectation from the calibration curve of proteins.

5. The authors claim that their signal is interferometric, so it should scale with the scattering amplitude rather than cross section. In this sense, it should scale identically to the scaling shown for iSCAT or mass photometry, which has been shown to be proportional to mass. Why is it different here, if we have essentially the same process going on (just a different detection methodology). If the signal is interferometric, why is always positive? Can it be negative?

Response:

We add the discussion at the 1st line after Equation (1). "The phase difference determines whether the interferometric contrast, namely the $2|E_o||E_s|\cos(\theta)$, is negative or positive. Unlike the interferometric scattering, where the Gouy phase shift dominates and leads to a negative interferometric contrast^{11,22},

the phase difference is about zero in ESM because of the short distance between scattering sites of surface roughness and analyte binding positions, leading to a positive interferometric contrast as the plasmonic scattering microscopy^{17, 18, 22, 23}.

The key point is that the phase determines whether the interferometric contrast is positive or negative, namely whether the spots are bright or dark. For the interferometric scattering (iSCAT) system, the Gouy phase shift brings the phase shift of π , making the image contrast negative. For the previously reported plasmonic scattering microscopy and evanescent scattering microscopy reported here, the proteins and surface roughness are very close to each other. Thus, the phases of their scattering are identical, resulting in positive image contrast. A group from Imperial College has theoretically proved this (ACS Photonics 8, 2227-2233 (2021); Physical Review Research 3, 033111 (2021)), and this has also been experimentally demonstrated by our evanescent plasmonic scattering microscopy (Nature Methods 17, 1010-1017 (2020); ACS Sensors 6, 1357-1366 (2021); ACS Sensors 6, 4244-4254 (2021)).

Regarding the positive and negative issues, we add representative line profiles of polystyrene nanoparticle and BSA protein in Supplementary Fig. N3.1 to show that the PSF of ESM is not always positive. The bright spot has asymmetrical intensity distribution, and some pixels included by the bright spot have negative intensities, similar to the results achieved by evanescent plasmonic scattering microscopy (Nature Methods 17, 1010-1017 (2020); ACS Sensors 6, 1357-1366 (2021)). This pattern is more complex than the Airy pattern, so we use mean intensity, rather than the maximum intensity, as the sensor output.

6. I very much struggle with the fundamental feasibility of their experiment. Using their equation for the scattering cross section, I obtain a cross section of about 10^{-11} um^2 for BSA (compare also with Piliarik and Sandoghdar Nature Communications 2014). This would amount to about 60 photons scattered per 5 ms in total, of which no more than 10% (and probably less with a 0.4 NA) would be detected in their approach (because most photons travel towards the higher index material. This would be impossible to detect even with the world's best camera (once distributed over a realistic PSF). According to these estimates, the authors would require about 50 times more power to record the kind of image of BSA that they show in Fig. 2. The images they show in Fig. 2 appear physically impossible to record.

Response:

We have added quantitative analysis in Supplementary Note 2 to convince the readers that we see single proteins.

We should point out that the evanescent field created by total internal reflection has 5 times intensity enhancement, and averaged period of 50 ms (10 frames) is employed for image processing. The feasibility of using evanescent field enhancement for improving signal-to-noise ratio has been demonstrated by the PSM (Nature Methods 17, 1010-1017 (2020)) and one iSCAT group from the US (Nat Commun 12, 1744 (2021); Nat Methods 18, 447 (2021)), and the approach of using image average to improve the signal-to-noise ratio is also widely used by iSCAT groups (Nat Commun 5, 4495 (2014); Science 360, 423-427 (2018)). These two factors combined can compensate for the 50 times higher power requirement estimated by the reviewer. We emphasize these two factors at the 2nd line of the 2nd paragraph in the 'ESM Setup and Imaging Principles' section.

We also present a quantitative analysis based on the Rayleigh scattering cross-section without considering the complex subwavelength properties of evanescent scattering in Supplementary Note 2. The quantitative analysis shows that the shot noise limited ESM measurement SNR is ~ 10 for single BSA proteins, agreeing with the experimental results.

Reviewer #3 (Remarks to the Author):

The paper has improved noticeably clarifying all the main issues. It is easier to read and more detailed explanations are given. All my queries were answered satisfactorily. Overall, the paper is reporting very interesting high-quality work. Yet still the writing should be improved. The paper still has many grammatical errors, and several sentences are not well constructed, making it harder to understand. Nevertheless, with some patience it can be understood.

After some English editing throughout, I recommend publication.

A note to the authors' response letter: Besides stating their response to each of the comments, in all cases, the authors should explain in detail the corresponding changes done the paper and pinpoint their location of such changes in the revised text; preferably reproducing the amended text just after their response to the reviewer. If no changes were done after a given comment, they should say so and justify it.

Response:

Thank you very much for the positive feedback on our work and thoughtful advice. We have revised some typos in the revised manuscript and supporting information, such as the wrong title at supporting information. We will pay more attention to improving the clearness of our response letter in the following work.

Reviewers' Comments:

Reviewer #1:

Remarks to the Author:

Point 1

The response is incorrect. Both in interferometric scattering and in evanescent scattering the reference and scattered waves originate from the same place. Interferometric scattering can produce both positive and negative contrast. It should also be noted that my question was not solely a question with respect to the sign, but also why the PSF seems to have changed dramatically from one iteration to another. In the previous iteration the spots were largely dark, but now the authors claim that this is impossible.

Point 4

I really do not know what to make of this statement. There are now tens if not hundreds of papers demonstrating that interferometric detection of proteins scales with mass, yet here non-quantitative statements about 'surface charge' somehow are used to account for huge differences in scattered light detection.

Point 5

This is the same as the response to point 1 and does not address my question.

Point to Point Response to Reviewer's Comments

We are grateful for the constructive feedback from the reviewers. Following the advice, we have added quantitative analysis, experimental results, and related references to address all the questions. The revisions on the manuscript and supplementary information are marked in blue color. We provide point-to-point responses after individual questions that are quoted in italic.

Reviewer #1:

Point 1

The response is incorrect. Both in interferometric scattering and in evanescent scattering the reference and scattered waves originate from the same place. Interferometric scattering can produce both positive and negative contrast. It should also be noted that my question was not solely a question with respect to the sign, but also why the PSF seems to have changed dramatically from one iteration to another. In the previous iteration the spots were largely dark, but now the authors claim that this is impossible.

Point 1 in last review: The authors have dropped their claim of detecting Protein A, owing due to low SNR. What is very surprising in the new manuscript is that all protein images in Fig2 and in the SI now appear to be perfect 'bright spots' almost like in fluorescence microscopy, as opposed to the interferometric PSFs we had in the past. How is that possible?

Response:

We are sorry that we did not explain this clearer in the last revision. We should point out that the PSF is always positive in our evanescent scattering system, and we have never stated it has negative contrast in any version of our manuscript. In the 1st version, we present the images after pseudo processing, where the reviewer may overlook the intensity bars that indicate the positive contrast of protein binding events, leading to the confusion described here. Therefore, we followed the reviewer's advice and changed to grayscale images since the first revision to clearly show the positive contrast of protein binding.

As we explained in the last revision, the reason for the positive PSF in the evanescent scattering system is because the phase of scattering light from objects and surface roughness are identical, which has been demonstrated theoretically (ACS Photonics 2021, 8, 2227-2233; Physical Review Research 2021, 3, 033111) and experimentally (Nature Methods 2020, 17, 1010-1017; ACS Sensors 2021, 6, 1357-1366).

Point 4

I really do not know what to make of this statement. There are now tens if not hundreds of papers demonstrating that interferometric detection of proteins scales with mass, yet here non-quantitative statements about 'surface charge' somehow are used to account for huge differences in scattered light detection.

Response:

We are sorry that we did not explain this clearer in the last revision. We have added new experimental data to address this question in revised Supplementary Note 2.

First, considering the refractive index difference between polystyrene and proteins, the polystyrene nanoparticle should provide ~82% interferometric signal intensity than the protein with the same

diameter based on the Rayleigh scattering model. Therefore, in the revised Supplementary Fig. N2.1, we excluded the polystyrene nanoparticle results and only used the proteins data for the linear fitting. This protein-only calibration curve shows that the polystyrene nanoparticles present only ~24% higher image intensity than the proteins with the same diameter.

Second, we checked our experimental conditions. In this paper, as described in the surface functionalization section in the Methods section, the polystyrene nanoparticles bind to the poly-L-lysine modified surface via electrostatic adsorption, and proteins bind to the carboxyl group modified surface via covalent bonding. The covalent bonding usually can ensure that the proteins stay tightly on the surface (J. Am. Chem. Soc. 2019, 141, 16071–16078), while the electrostatic absorption may just loosely absorb the objects, leading to smaller image intensity due to the exponentially decaying property of evanescent fields (Anal. Chem. 2010, 82, 234–240). Considering that antibody modified nanoparticles usually have complicated behaviors on the surface (J. Am. Chem. Soc. 2019, 141, 16071–16078), the ESM image intensities of proteins on the poly-L-lysine modified surface were measured for comparison with those of 27.9 nm polystyrene nanoparticles (Supplementary Fig. N2.2). The experimental results show that the polystyrene nanoparticles present ~80% higher ESM image intensity than the proteins with the same diameter under the same binding conditions, agreeing with the theoretical predictions.

Point 5

This is the same as the response to point 1 and does not address my question.

Point 5 in last review: The authors claim that their signal is interferometric, so it should scale with the scattering amplitude rather than cross section. In this sense, it should scale identically to the scaling shown for iSCAT or mass photometry, which has been shown to be proportional to mass. Why is it different here, if we have essentially the same process going on (just a different detection methodology). If the signal is interferometric, why is always positive? Can it be negative?

Response:

We added Supplementary Fig. N2.3 to show that the ESM image intensity is proportional to the mass and does not conflict with the iSCAT and mass photometry results. As described in iSCAT literature (Nat Commun 2014, 5, 4495), the interferometric contrast scale with the molecular weight because the signal is proportional to the protein polarizability and therefore its volume. The volume is linearly related to the cube of the diameter. Thus, the cubic power law of d^3 , where d is the object diameter, achieved in Fig. 2 does not conflict with the results of iSCAT or mass photometry. In this paper, we use diameter for easy comparison of the measurement results of ESM with dynamic light scattering.

In addition, we have not found the negative signal for binding events in evanescent scattering system, including ESM and previously reported plasmonic scattering microscopy (Nature Methods 2020, 17, 1010-1017; ACS Sensors 2021, 6, 1357-1366). The iSCAT can achieve negative contrast on the focus position, and the positive contrast by moving the objective focus plane or sample position by hundreds of nanometers (Optics Express 2020, 28, 25969-25988). However, the evanescent field created by total internal reflection or surface plasmon resonance is limited to ~100 nanometers away from the sensor surface. Thus, we can only achieve the positive image contrast for the binding events because the phase of scattering light from objects and surface roughness are identical, which has been demonstrated theoretically (ACS Photonics 2021, 8, 2227-2233; Physical Review Research 2021, 3, 033111) and experimentally (Nature Methods 2020, 17, 1010-1017; ACS Sensors 2021, 6, 1357-1366).

Reviewers' Comments:

Reviewer #1:

Remarks to the Author:

I have no further comments